# Fusion of Identification Information from ESM Sensors and Radars Using Dezert–Smarandache Theory Rules

**Tadeusz Pietkiewicz** 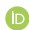

Institute of Radioelectronics, Faculty of Electronics, Military University of Technology, 00-908 Warsaw, Poland; tadeusz.pietkiewicz@wat.edu.pl

**Abstract:** This paper presents a method of fusion of identification (attribute) information provided by two types of sensors: combined primary and secondary (IFF) surveillance radars and ESMs (electronic support measures). In the first section, the basic taxonomy of attribute identification is adopted in accordance with the standards of STANAG 1241 ed. 5 and STANAG 1241 ed. 6 (draft). These standards provide the following basic values of the attribute identifications: FRIEND; HOSTILE; NEUTRAL; UNKNOWN; and additional values, namely ASSUMED FRIEND and SUSPECT. The basis of theoretical considerations is Dezert–Smarandache theory (DSmT) of inference. This paper presents and uses in practice six information-fusion rules proposed by DSmT, i.e., the proportional conflict redistribution rules (PCR1, PCR2, PCR3, PCR4, PCR5, and PCR6), for combining identification information from different ESM sensors and radars. This paper demonstrates the rules of determining attribute information by an ESM sensor equipped with the database of radar emitters. It is proposed that each signal vector sent by the ESM sensor contains an extension specifying a randomized identification declaration (hypothesis)—a basic belief assignment (BBA). This paper also presents a model for determining the basic belief assignment for a combined primary and secondary radar. Results of the PCR rules of sensor information combining for different scenarios of a radio electronic situation (deterministic and Monte Carlo) are presented in the final part of this paper. They confirm the legitimacy of the use of Dezert–Smarandache theory in information fusion for primary radars, secondary radars, and ESM sensors.

**Keywords:** information fusion; Dezert–Smarandache theory (DSmT) of inference; conflict redistribution rules; radar emitters recognition; electronic support measures (ESMs); primary and secondary radars

## 1. Introduction

Designing systems for creating a recognized air picture in the air defense system requires, among other factors, the development of algorithms for combining identification information about detected targets from various types of sensors. The basic elements of the air-situation-recognition system are two types of sensors: ESM sensors and combined primary and secondary radars (IFF: identification friend or foe). They provide identification information to information-processing centers that develop a recognized air picture. Each air-situation information-processing system should have an attribute information set, specifying acceptable values for the identification information of the detected targets transmitted by sensors to information-processing centers and information produced by those centers. This article uses a certain interpretation of attribute identification in accordance with the NATO STANAG 1241 standard [1,2]. It should be noted that this is one of the possible interpretations of the adopted definitions. It is assumed that five identification classes are used: three primary and two secondary ones. Sensors can transmit identification information in the form of a hard decision, sometimes determined as non-randomized, or a soft decision, sometimes determined as a randomized decision. In this paper, it is assumed that the sensors send identification information to the system in a randomized form, i.e., in the form of a basic belief assignment on the set of identification classes. This

assignment determines the sensor's belief that the detected emitter belongs to separate identification classes.

Another problem that should be solved by the designers of air-situation-recognition systems is the choice of the method of combining information from sensors. One possible solution is the STANAG 4162 standard proposed by NATO. It is a standard based on Bayesian decision functions. Another solution is to use Dempster–Shafer reasoning [3,4]. Works [5,6] show that such solutions have their drawbacks. The disadvantages of Dempster–Shafer reasoning are also confirmed in this work for situations of high conflict. These disadvantages can be avoided by applying Dezert–Smarandache theory (DSmT).

DSmT in its basic version contains five rules of proportional conflict redistribution, namely PCR1-PCR5. Later, the authors presented a new PCR6 rule [7], which was tested in this paper, alongside the earlier PCR1-PCR5 rules.

At this point, it should be noted that DSmT is not the only development of Dempster–Shafer theory. DSmT provides different rules for the proportional redistribution of the conflict mass between the resulting BBA masses in the process of information fusion. Examples of this development of Dempster–Shafer theory can be found in [8–10]. In [8,9], it was proposed to use the negation evidence (BBAs from sensors) and Deng's [11] entropy to determine BBAs after information fusion. The new BBA distribution is calculated as a weighted sum of the original BBA, with the weights being a function of Deng's entropy. The paper [10] shows the application of the negation evidence method in fault diagnosis.

In [12], a new risk priority model based on the belief Jensen–Shannon divergence measure and Deng's entropy is proposed. In the new method, Deng's entropy and the belief Jensen–Shannon divergence measure are used to model the uncertainty of risk assessments in the "Failure mode and effects analysis" procedure and to deal with potential conflict information. This allows one to calculate the appropriate weighted average probabilities (WAPs) value. Classic Dempster's combination rule is used to fuse data to generate integrated values of the risk factors. Unfortunately, the latest publications do not compare the DSmT method with the proposed new solutions. The examples provided there concern other applications than those presented in this work.

In [13], a generalized evidential Jensen–Shannon (GEJS) divergence to measure the conflict and disparity among multiple sources of evidence. This generalization was used to determine the weights of the information sources. Subsequently, it was used to determine the results of information fusion using Dempster's combination rule.

In [14], an extension of Dempster–Shafer theory was presented, which received the name complex evidence theory. It implements complex weighted discounting multisource information fusion. The complex evidence theory defines complex basic belief assignments and a complex evidential correlation coefficient. A weighted discounting multisource information-fusion algorithm with complex evidential correlation coefficient improves the performance of expert systems based on complex evidence theory.

Another development of Dempster–Shafer theory is the generalized quantum evidence theory [15,16]. In these papers, multisource quantum information fusion was presented. The papers are complemented by an example of a pattern classifier from the motor-rotor-fault-diagnosis domain, which confirmed the efficiency of the multisource quantum information-fusion algorithm.

This paper is a continuation of another work [17] and contains new research results obtained for new scenarios of the electronic situation, new DSmT rules, and new information-fusion schemes.

Below, the substantive content of individual chapters is subsequently discussed. The first part of the paper presents the applied interpretation of attribute identification in accordance with the NATO STANAG 1241 standard. It should be noted that this is one of the possible interpretations of the adopted definitions. It leads to the Bayesian model of the basic belief assignment.

The next part of the paper presents the mathematical form of the DSmT rules PCR1, PCR2, PCR3, PCR4, PCR5, and PCR6 [5,18] for two sensor inputs and PCR5 and PCR6

for three sensor inputs, assuming the Bayesian model of the basic belief assignment of the hypothesis.

The next two sections, namely Sections 4 and 5, show how to determine the basic belief assignment for a combined primary and secondary (IFF) radar and ESM sensors. These assignments are the input information in the PCR1-PCR6 information-fusion algorithms. Section 4 presents a method for determining the basic belief assignments (BBAs) of airborne targets moving in the observation space of a combined primary and secondary (IFF) radar sensor. This method uses the primary radar model, taken from [17]. The result of executing the algorithm of this method are scenarios containing reports from BBAs. Section 5 presents a method for determining BBAs for airborne targets that emit electromagnetic radiation (airborne radars and other emitters). It requires databases of reference signals of various airborne emitters, equipment of airborne platforms, and the nationality of the platforms.

Each sensor report sent to the information-fusion center contains a vector of belief mass for all attribute identification values. The results of the proportional conflict redistribution sensor information, combining rules for selected deterministic and Monte Carlo scenarios, are presented in Sections 6 and 7 of the paper. Section 6 presents the results of research on the fusion of information sent only from ESM sensors. This corresponds to a situation when the ESM sensors operate in a system: one master station and one or two slave stations. Section 7 presents the results of research on the fusion of information sent from ESM sensors and combined primary and secondary radars. This corresponds to a situation where one ESM sensor (master station) and radars cooperate with the information-processing center (the producer of the recognized air picture).

Conclusions are provided at the end of the paper. They confirm the legitimacy of the use of DSmT in information fusion for primary radars, secondary radars, and ESM sensors.

## 2. Interpretation of Attribute Identification according to STANAG 1241

The set of possible values of attribute identifications used by sensors can be adopted based on standardization documents of organizations that exploit these sensors [1,2,19–21].

This paper assumes a basic taxonomy of identification in accordance with the draft of STANAG 1241 ed. 6 [2]. Other similar documents may include the following standards: STANAG 4420 and STANAG 1241 ed. 5, which provide the following basic values of the attribute identifications:

- FRIEND (F);
- HOSTILE (H);
- NEUTRAL (N);
- UNKNOWN (U).

Each of these documents contain their own definitions of the declarations.

The following definitions of these basic values of the attribute identification are used in this paper (in accordance with [2]):

- FRIEND: an allied/coalition military track, object, or entity; a track, object, or entity, supporting friendly forces and belonging to an allied/coalition nation or a declared or recognized friendly faction or group;
- HOSTILE: a track, object, or entity whose characteristics, behavior, or origin indicate that it belongs to opposing forces, or that it poses a threat to friendly forces or their mission;
- NEUTRAL: a military or civilian track, object, or entity, neither belonging to allied/coalition military forces nor to opposing military forces, whose characteristics, behavior, origin, or nationality indicate that it is neither supporting nor opposing friendly forces or their mission;
- UNKNOWN: an evaluated track, object, or entity, which do not meet the criteria for any other standard identity.

These standards bring additional values of the attribute identification:

- ASSUMED FRIEND (AF);
- SUSPECT (S).

Attention should be paid to these two last identities contained in [1], as well as their definitions [2]:

- ASSUMED FRIEND: a track, object, or entity, which is assumed to be friend or neutral because of its characteristics, behavior, or origin;
- SUSPECT: a track, object, or entity whose characteristics, behavior, or origin indicate that it potentially belongs to opposing forces or potentially poses a threat to friendly forces or their mission.

The identification definitions in [1,2] can lead to different interpretations. This paper adopts the interpretation that is shown by the graphical form of in Figure 1.

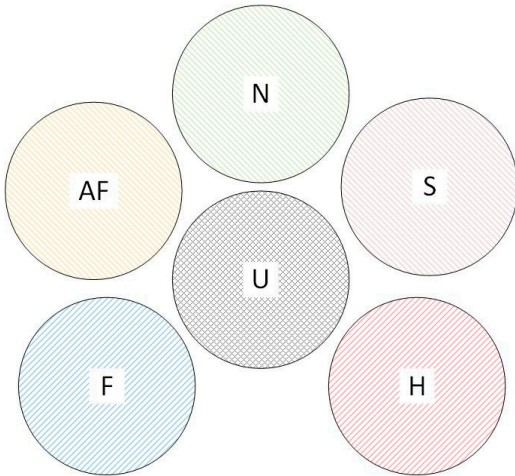

**Figure 1.** The interpretation of STANAG 1241 using the Venn diagram.

### 3. Fusion of Information from ESM Sensors and Radars in the Information-Fusion Center (IFC)

*3.1. Diagram of the Process of Information Fusion for Two Sensors in the Information-Fusion Center*

In this work, it is assumed that ESM sensors send messages asynchronously to the information-fusion center. These reports contain sensor decisions regarding the identification of objects emitting the detected signals. The set of possible identifications is as follows:

$$\Theta = \{\theta_i, = 1, \ldots, 6\}, \tag{1}$$

where in the following interpretation is used:

$\theta_1$: FRIEND (F);
$\theta_2$: HOSTILE (H);
$\theta_3$: NEUTRAL (N);
$\theta_4$: ASSUMED FRIEND(AF);
$\theta_5$: SUSPECT(S);
$\theta_6$: UNKNOWN (U).

According to Figure 1, the hypotheses are mutually exclusive, i.e.,

$$\theta_i \cap \theta_j = \begin{cases} \theta_i, & \text{if } i = j, \\ \varnothing, & \text{if } i \neq j. \end{cases} \tag{2}$$

Each sensor with the number $i$ ($i \in \mathbb{N}$) sends its decisions as so-called soft decisions, i.e., as BBA measure vectors (BBA: basic belief assignment).

$$\boldsymbol{m}_i = [m_i(\theta_1), \ldots, m_i(\theta_6)]. \tag{3}$$

A vector of generalized BBA measures for the information-fusion center should also be introduced as follows:

$$\boldsymbol{m}_F = [m_F(\theta_1), \ldots, m_F(\theta_6)]. \tag{4}$$

This paper adopts the Bayesian BBA model as it has been adopted as valid in the STANAG 4162 standard [20]. This means that Equation (5) applies in addition to (1) and (2).

$$\sum_{i=1}^{6} m_F(\theta_i) = \sum_{i=1}^{6} m_i(\theta_i) = 1. \tag{5}$$

In the first case that is considered, two sensors send, asynchronously in one cycle, one report each, containing decisions regarding the BBAs related to the target. The IFC system, after receiving the report from the sensor, fuses the information contained in the two vectors: in the current generalized BBA vector $\boldsymbol{m}_F = [m_F(\theta_1), \ldots, m_F(\theta_6)]$ and in the BBA vector $\boldsymbol{m}_1$ from sensor 1 or in the BBA vector $\boldsymbol{m}_2$ from sensor 2.

The information-fusion procedure performed in the IFC is carried out in accordance with the following formula:

$$\boldsymbol{m}_F^{'} = R_F(\boldsymbol{m}_F, \boldsymbol{m}_i) \ (i = 1 \text{ or } 2), \tag{6}$$

wherein $\boldsymbol{m}_F^{'}$ is a vector of the generalized BBA measure determined by the $R_F$ rule based on the previous generalized BBA measure vector $\boldsymbol{m}_F$ and the new BBA measure vector $\boldsymbol{m}_i$ sent by the $i$-th sensor. The diagram of identification information fusion from the ESM sensors is shown in Figure 2.

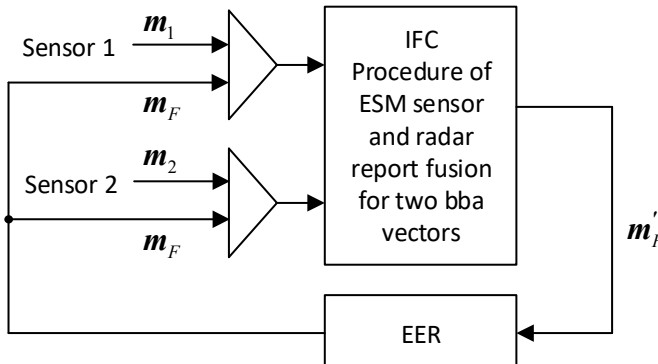

**Figure 2.** The diagram of the information-fusion process in the information-fusion center (IFC) for two sensors. Explanations: $m_i$ a BBA measure vector of $i$-th sensor; $m_F$, a generalized BBA measure vector that is a part of the electronic entity record in the IFC; EER, an electronic entity record in the IFC database.

In the second case that is considered, two sensors send, asynchronously in one cycle, one report each, containing decisions regarding the BBAs related to the target. The IFC system waits for reports from both sensors in one cycle, using registers. Only when both registers are full does the IFC system perform a fusion of the information contained in three vectors: BBA vector $\boldsymbol{m}_F = [m_F(\theta_1), \ldots, m_F(\theta_6)]$, BBA vector $\boldsymbol{m}_1$ from sensor 1, and BBA vector $\boldsymbol{m}_2$ from sensor 2. It should be noted that this method has a drawback: the information stored in the registers lose credibility.

In this case, the information-fusion procedure performed in the IFC is carried out in accordance with the following formula:

$$\boldsymbol{m}_F^{'} = R_F(\boldsymbol{m}_F, \boldsymbol{m}_1, \boldsymbol{m}_2), \tag{7}$$

wherein $\boldsymbol{m}_F^{'}$ is a vector of the generalized BBA measure determined by the $R_F$ rule based on the previous generalized BBA measure vector $\boldsymbol{m}_F$ and the new BBA measure vectors $\boldsymbol{m}_1$ and $\boldsymbol{m}_2$ sent by both sensors. The diagram of identification information fusion from the ESM sensors is shown in Figure 3.

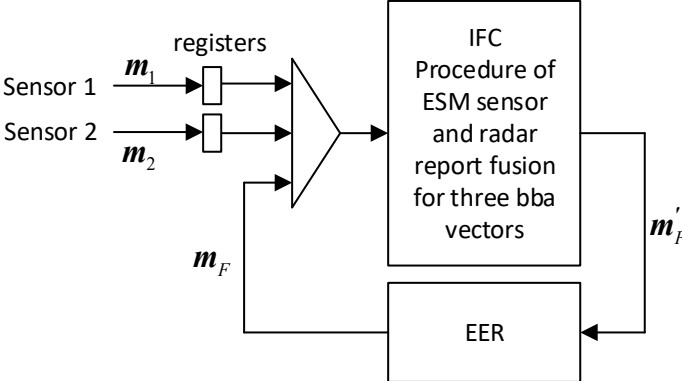

**Figure 3.** The diagram of the information-fusion process in the information-fusion center (IFC) for two sensors and electronic entity record from the IFC database. Explanations: $m_i$, a BBA measure vector of *i*-th sensor; $m_F$, a generalized BBA measure vector that is a part of the electronic entity record in the IFC; EER, an electronic entity record in the IFC database.

Further in this paper, the combination rules of the BBA vector from the *i*-th sensor and the generalized BBA vector in the CFI are described.

### 3.2. The Rules of Combination of BBA Measures Vectors

This section presents formulas defining various combination rules for calculating basic belief assignments for the system shown in Figures 2 and 3. The general forms are described in detail in [6,18,22]. The information-fusion rules of the DSm theory are presented below with the following constraints:

- the properties of a set of hypotheses are described by Formulas (1) and (2);
- for the first scheme (Figure 2), the information-fusion procedure handles two information inputs: on one input, reports from two ESM sensors appear alternately, while on the second input, electronic entity records from the IFC database appear;
- for the second scheme (Figure 3), the information-fusion procedure handles three information inputs: on the first input, reports from a combined primary and secondary surveillance radar appear; on the second input, reports from an ESM sensor appear; and on the third input, electronic entity records from the IFC database appear.

#### 3.2.1. Dempster's Rule

Dempster's rule [3,4] of the BBA measure vector $m_i$ sent by the *i*-th sensor and the generalized BBA measure vector $m_F$ s in the IFC is described for each $\theta_j \in \Theta$ by the following formula:

$$m_F'(\theta_j) = m_D(\theta_j) = \frac{\sum\limits_{\substack{k=1,\dots,6 \\ l=1,\dots,6 \\ \theta_k \cap \theta_l = \theta_j}} m_F(\theta_k)m_i(\theta_l)}{1 - \sum\limits_{\substack{k=1,\dots,6 \\ l=1,\dots,6 \\ \theta_j \cap \theta_k = \varnothing}} m_F(\theta_k)m_i(\theta_l)} =$$

$$= \frac{m_F(\theta_j)m_i(\theta_j)}{1 - \sum_{k=1}^{6}\sum\limits_{\substack{l=1 \\ l \neq k}}^{6} m_F(\theta_k)m_i(\theta_l)} = \frac{m_{Fi}(\theta_j)}{1 - k_{Fi}} , \tag{8}$$

where in the $k_{Fi}$ degree of conflict is defined by the formula:

$$k_{Fi} = \sum\limits_{\substack{k=1,\dots,6 \\ l=1,\dots,6 \\ \theta_j \cap \theta_k = \varnothing}} m_F(\theta_k)m_i(\theta_l) = \sum_{k=1}^{6}\sum\limits_{\substack{l=1 \\ l \neq k}}^{6} m_F(\theta_k)m_i(\theta_l) , \tag{9}$$

while

$$m_{Fi}(\theta_j) = m_F(\theta_j) m_i(\theta_j). \tag{10}$$

It can be noticed that

$$\sum_{\substack{k=1,\dots,6 \\ l=1,\dots,6}} m_F(\theta_k) m_i(\theta_l) = 1 \,, \tag{11}$$

$$\begin{aligned}
\sum_{\substack{k=1,\dots,6 \\ l=1,\dots,6 \\ \theta_j \cap \theta_k = \varnothing}} m_F(\theta_k) m_i(\theta_l) &+ \sum_{\substack{k=1,\dots,6 \\ l=1,\dots,6 \\ \theta_j \cap \theta_k \neq \varnothing}} m_F(\theta_k) m_i(\theta_l) = \\
&= \sum_{k=1}^{6} \sum_{\substack{l=1 \\ l \neq k}}^{6} m_F(\theta_k) m_i(\theta_l) + \sum_{k=1}^{6} \sum_{\substack{l=1 \\ l=k}}^{6} m_F(\theta_k) m_i(\theta_l) = \\
&= \sum_{k=1}^{6} \sum_{\substack{l=1 \\ l \neq k}}^{6} m_F(\theta_k) m_i(\theta_l) + \sum_{k=1}^{6} m_F(\theta_k) m_i(\theta_k) = 1 \,.
\end{aligned} \tag{12}$$

From (12), it follows that if

$$\sum_{k=1}^{6} m_F(\theta_k) m_i(\theta_k) = 1 \,, \text{i.e.,} \sum_{\substack{k=1,\dots,6 \\ l=1,\dots,6}} m_F(\theta_k) m_i(\theta_l) = 0 \,, \tag{13}$$

then the degree of conflict is full.

If

$$\sum_{k=1}^{6} m_F(\theta_k) m_i(\theta_k) = 0 \,, \text{i.e.,} \sum_{\substack{k=1,\dots,6 \\ l=1,\dots,6}} m_F(\theta_k) m_i(\theta_l) = 1 \,, \tag{14}$$

then there is no conflict.

$m_D(.)$ is the Dempster–Shafer fusion result if, and only if, the denominator of expression (8) is non-zero, i.e., if the degree of conflict $k_{Fi}$ is less than 1.

### 3.2.2. The Proportional Conflict Redistribution Rule PCR1

The PCR1 rule is the simplest and the easiest version of the proportional conflict redistribution rule. The concept of the PCR1 rule assumes the calculation of the total conflicting mass (not worrying about the partial conflicting masses). The total conflicting mass is redistributed to all non-empty sets of hypotheses proportionally, with respect to their corresponding non-empty column sum of the associated mass matrix. The PCR1 rule is defined for every non-empty hypothesis in the following way:

$$\begin{aligned}
m'_F(\theta_j) = m_{PCR1}(\theta_j) &= \Big[ \sum_{\substack{k=1,\dots,6 \\ l=1,\dots,6 \\ \theta_k \cap \theta_l = \theta_j}} m_F(\theta_k) m_i(\theta_l) \Big] + \frac{c_{Fi}(\theta_j)}{d_{Fi}} \cdot k_{Fi} = \\
&= m_F(\theta_j) m_i(\theta_j) + \frac{c_{Fi}(\theta_j)}{d_{Fi}} \cdot k_{Fi} = m_{Fi}(\theta_j) + \frac{c_{Fi}(\theta_j)}{d_{Fi}} \cdot k_{Fi}
\end{aligned} \tag{15}$$

where $c_{Fi}(\theta_j)$ is the non-zero sum of the column corresponding to the hypotheses $\theta_j$ in the mass matrix

$$M = \begin{bmatrix} m_F \\ m_i \end{bmatrix} \tag{16}$$

specified by the following formula:

$$c_{Fi}(\theta_j) = m_F(\theta_j) + m_i(\theta_j) \tag{17}$$

where:

- $m_i$ (i = 1,2) is a row vector of the basic belief assignment masses of the *i*-th sensor's hypotheses;
- $m_F$ is a row vector of the basic belief assignments masses of the IFC system's hypotheses;
- $k_{Fi}$ is the degree of mass conflict specified by the following formula:

$$k_{Fi} = \sum_{\substack{k=1,\dots,6 \\ l=1,\dots,6 \\ \theta_k \cap \theta_l = \varnothing}} m_F(\theta_k) m_i(\theta_l) = \sum_{k=1}^{6} \sum_{\substack{l=1 \\ l \neq k}}^{6} m_F(\theta_k) m_i(\theta_l), \qquad (18)$$

- $d_{Fi}$ is the sum of all non-zero column sums of all non-empty sets, as follows:

$$d_{Fi} = \sum_{j=1}^{6} \left[ m_F(\theta_j) + m_i(\theta_j) \right] = \sum_{j=1}^{6} c_{Fi}(\theta_j) . \qquad (19)$$

In the case from this paper, $d_{Fi} = 2$ because

$$\sum_{j=1}^{6} m_F(\theta_j) = \sum_{j=1}^{6} m_i(\theta_j) = 1 \qquad (20)$$

In addition,

$$m_{Fi}(\theta_j) = m_F(\theta_j) m_i(\theta_j) \qquad (21)$$

3.2.3. The Proportional Conflict Redistribution Rule PCR2

In the PCR2 rule, the total conflicting mass $k_{Fi}$ is distributed only to the non-empty sets involved in the conflict (not to all non-empty sets) and taken proportionally with respect to their corresponding non-empty column sum.

A non-empty set $\theta_k \in \Theta$ is considered to be involved in the conflict if there exists another set $\theta_l \in \Theta$ that is neither included in $\theta_k$ nor includes a $\theta_k$ such value that $\theta_k \cap \theta_l = \varnothing$ and $m_{Fi}(\theta_k \cap \theta_l) > 0$. The PCR2 rule is defined for every non-empty hypothesis $\theta_j \in \Theta$ in the following way:

$$m'_F(\theta_j) = m_{PCR2}(\theta_j) = \left[ \sum_{\substack{k=1,\dots,6 \\ l=1,\dots,6 \\ \theta_k \cap \theta_l = \theta_j}} m_F(\theta_k) m_i(\theta_l) \right] + C(\theta_j) \frac{c_{Fi}(\theta_j)}{e_{Fi}} \cdot k_{Fi} =$$

$$= m_F(\theta_j) m_i(\theta_j) + C(\theta_j) \frac{c_{Fi}(\theta_j)}{e_{Fi}} \cdot k_{Fi} = m_{Fi}(\theta_j) + C(\theta_j) \frac{c_{Fi}(\theta_j)}{e_{Fi}} \cdot k_{Fi} \qquad (22)$$

where

$$C(\theta_j) = \begin{cases} 1, & \text{if } \theta_j \text{ is involved in the conflict,} \\ 0, & \text{otherwise.} \end{cases} \qquad (23)$$

Formula (23) can be written differently in form (25), taking into account the definition of the involvement in a conflict and Formula (24) [6]:

$$m_{Fi}(\theta_j \cap \theta_k) = m_F(\theta_j) \cdot m_i(\theta_k) + m_F(\theta_k) \cdot m_i(\theta_j) \qquad (24)$$

$$C(\theta_j) = \begin{cases} 1, & \text{if } \exists \, \theta_k \in \Theta, k \neq j : m_F(\theta_j) \cdot m_i(\theta_k) + m_F(\theta_k) \cdot m_i(\theta_j) > 0 \\ 0, & \text{otherwise.} \end{cases} \qquad (25)$$

$c_{Fi}(\theta_j)$ is the non-zero sum of the column corresponding to the hypotheses in the mass matrix M (16) specified by the following formula:

$$c_{Fi}(\theta_j) = m_F(\theta_j) + m_i(\theta_j) \qquad (26)$$

where:

- $m_i$ (i = 1,2) is a row vector of the basic belief assignment; masses of the *i*-th sensor's hypotheses;
- $m_F$ is a row vector of the basic belief assignment masses of the IFC system's hypotheses;
- $k_{Fi}$ is the degree of mass conflict specified by Formula (18);
- $e_{Fi}$ is the sum of all non-zero column sums of all non-empty sets involved in the conflict, as follows:

$$e_{Fi} = \sum_{j \in CF} [m_F(\theta_j) + m_i(\theta_j)] = \sum_{j \in CF} c_{Fi}(\theta_j) = \sum_{j=1}^{6} C(\theta_j)[m_F(\theta_j) + m_i(\theta_j)] = \sum_{j=1}^{6} C(\theta_j) \cdot c_{Fi}(\theta_j) \tag{27}$$

where

$$CF = \{j = 1, ..., 6 : \forall \theta_k \in \Theta \ m_{Fi}(\theta_j \cap \theta_k) > 0\} \tag{28}$$

and $m_{Fi}(\theta_j \cap \theta_k)$ is defined by (24).

In addition

$$m_{Fi}(\theta_j) = m_F(\theta_j) m_i(\theta_j) \tag{29}$$

It is shown below that in the case of data used in numerical experiments (Section 6), $e_{Fi} = 2$, which means that the PCR2 rule is equivalent to the PCR1 rule. The BBA vectors used there contain values less than 1, which means the following:

$$\forall j = 1, \dots, 6 : m_F(\theta_j) < 1 \land m_i(\theta_j) < 1 \tag{30}$$

It follows that each BBA vector contains at least two non-zero components, that is $\exists j = 1, \dots, 6, \ \exists k = 1, \dots, 6$ with $k \neq j$, such that

$$0 < m_F(\theta_j) < 1 \land 0 < m_F(\theta_k) < 1 \tag{31}$$

$$0 < m_i(\theta_j) < 1 \land 0 < m_i(\theta_k) < 1 \tag{32}$$

From (31) and (32), it follows that if $m_F(\theta_j) > 0$, then there exists at least one value $k \neq j$, such that $m_i(\theta_k) > 0$, which can be written in the following form:

$$\forall j = 1, \dots, 6 : \ 0 < m_F(\theta_j) < 1 \ \Rightarrow \ \exists k \dots \dots, 6, k \neq j : \ 0 < m_i(\theta_k) < 1 \tag{33}$$

From (33), it follows that

$$\forall j = 1, \dots, 6 : \ 0 < m_F(\theta_j) < 1 \ \Rightarrow \ \exists k = 1, \dots, 6, k \neq j : \ m_F(\theta_j) \cdot m_i(\theta_k) > 0 \tag{34}$$

The same applies to the following:

$$\forall j = 1, \dots, 6 : \ 0 < m_i(\theta_j) < 1 \ \Rightarrow \ \exists k = 1, \dots, 6, k \neq j : \ m_i(\theta_j) \cdot m_F(\theta_k) > 0 \tag{35}$$

Taking into account (34), (35) and (25) can obtain the following:

$$\forall j = 1, \dots, 6 : \ 0 < m_F(\theta_j) < 1 \ \Rightarrow \ \exists k = 1, \dots, 6, k \neq j \text{ such that}$$
$$m_F(\theta_j) \cdot m_i(\theta_k) + m_i(\theta_j) \cdot m_F(\theta_k) > 0 \tag{36}$$

$$\forall j = 1, \dots, 6 : \ 0 < m_i(\theta_j) < 1 \ \Rightarrow \ \exists k = 1, \dots, 6, k \neq j \text{ such that}$$
$$m_i(\theta_j) \cdot m_F(\theta_k) + m_F(\theta_j) \cdot m_i(\theta_k) > 0 \tag{37}$$

From (36) and (37), it follows that

$$\forall j = 1, \dots, 6 : \ 0 < m_F(\theta_j) < 1 \ \Rightarrow \ C(\theta_j) = 1 \tag{38}$$

$$\forall j = 1, \dots, 6 : \ 0 < m_i(\theta_j) < 1 \ \Rightarrow \ C(\theta_j) = 1 \tag{39}$$

This means that any hypothesis with a non-zero BBA value for any of the two sensors is involved in a conflict.

From (27), it follows that

$$e_{Fi} = \sum_{j=1}^{6} C(\theta_j)\left[m_F(\theta_j) + m_i(\theta_j)\right] = \sum_{j=1}^{6} C(\theta_j)m_F(\theta_j) + \sum_{j=1}^{6} C(\theta_j)m_i(\theta_j) \qquad (40)$$

Using (36), (37) and (40), the value $e_{Fi}$ is determined.

Because

$$\sum_{j=1}^{6} C(\theta_j)m_F(\theta_j) = \sum_{\substack{j=1 \\ m_F(\theta_j)>0}}^{6} C(\theta_j)m_F(\theta_j) + \sum_{\substack{j=1 \\ m_F(\theta_j)=0}}^{6} C(\theta_j)m_F(\theta_j) = \sum_{\substack{j=1 \\ m_F(\theta_j)>0}}^{6} m_F(\theta_j) = 1 \quad (41)$$

$$\sum_{j=1}^{6} C(\theta_j)m_i(\theta_j) = \sum_{\substack{j=1 \\ m_i(\theta_j)>0}}^{6} C(\theta_j)m_i(\theta_j) + \sum_{\substack{j=1 \\ m_i(\theta_j)=0}}^{6} C(\theta_j)m_i(\theta_j) = \sum_{\substack{j=1 \\ m_i(\theta_j)>0}}^{6} m_i(\theta_j) = 1 \quad (42)$$

we obtain

$$e_{Fi} = \sum_{j=1}^{6} C(\theta_j)m_F(\theta_j) + \sum_{j=1}^{6} C(\theta_j)m_i(\theta_j) = 2 \qquad (43)$$

Considering (43), it can be said that in this case, the PCR2 rule is equivalent to the PCR1 rule. For this reason, the results of the PCR2 rule are not presented in Section 6, as they would be identical to the results of the PCR1 rule as only the Bayesian BBAs are used in this application.

### 3.2.4. The Proportional Conflict Redistribution Rule PCR3

In the PCR3 rule, the partial conflicting masses are distributed instead of the total conflicting mass, $k_{Fi}$, to the non-empty sets involved in the partial conflict. If an intersection is empty, for instance $\theta_k \cap \theta_l = \varnothing$, then the mass $m(\theta_k \cap \theta_l)$ of the partial conflict is transferred to the non-empty sets $\theta_k$ and $\theta_l$ proportionally, with respect to the non-zero sum of masses assigned to $\theta_k$ and, respectively, to $\theta_l$ by BBAs $m_F(.)$ and $m_i(.)$. The PCR3 rule works if at least one set between $\theta_k$ and $\theta_l$ is non-empty and its column sum is non-zero.

The PCR3 rule is defined for every non-empty hypothesis $\theta_j \in \Theta$ in the following way:

$$m_F'(\theta_j) = m_{PCR3}(\theta_j) = \left[\sum_{\substack{k=1,\dots,6 \\ l=1,\dots,6 \\ \theta_k \cap \theta_l = \theta_j}} m_F(\theta_k)m_i(\theta_l)\right] + \left[c_{Fi}(\theta_j)\sum_{\substack{k=1,\dots,6 \\ \theta_k \cap \theta_j = \varnothing}} S_{Fi}^{PCR3}(\theta_j,\theta_k)\right] =$$

$$= m_{Fi}(\theta_j) + \left[c_{Fi}(\theta_j)\sum_{\substack{k=1,\dots,6 \\ k \neq j}} S_{Fi}^{PCR3}(\theta_j,\theta_k)\right]$$

$$(44)$$

where

$$S_{Fi}^{PCR3}(\theta_j,\theta_k) = \begin{cases} \dfrac{m_F(\theta_k)m_i(\theta_j)+m_F(\theta_j)m_i(\theta_k)}{c_{Fi}(\theta_j)+c_{Fi}(\theta_k)} & \text{for } c_{Fi}(\theta_j)+c_{Fi}(\theta_k) \neq 0 \\ 0 \text{ for } c_{Fi}(\theta_j)+c_{Fi}(\theta_k) = 0 \end{cases} \qquad (45)$$

$c_{Fi}(\theta_j)$ is the non-zero sum of the column corresponding to the hypotheses $\theta_j$ in the mass matrix $\boldsymbol{M}$ (16), specified by the following formula:

$$c_{Fi}(\theta_j) = m_F(\theta_j) + m_i(\theta_j) \tag{46}$$

### 3.2.5. The Proportional Conflict Redistribution Rule PCR4

The PCR4 rule redistributes the partial conflicting masses only to the sets involved in the partial conflict in proportion to the non-zero mass sum assigned to $\theta_k$ and $\theta_l$ by the conjunction rule according to the following formula:

$$
\begin{aligned}
m'_F(\theta_j) = m_{PCR4}(\theta_j) &= m_{Fi}(\theta_j) + m_{Fi}(\theta_j) \sum_{\substack{k=1,\ldots,6 \\ \theta_k \cap \theta_j = \varnothing}} S_{Fi}^{PCR4}(\theta_j, \theta_k) = \\
&= m_{Fi}(\theta_j) + m_{Fi}(\theta_j) \sum_{\substack{k=1,\ldots,6 \\ k \neq j}} S_{Fi}^{PCR4}(\theta_j, \theta_k)
\end{aligned}
\tag{47}
$$

where

$$
S_{Fi}^{PCR4}(\theta_j, \theta_k) = 
\begin{cases}
\frac{m_{Fi}(\theta_j \cap \theta_k)}{m_{Fi}(\theta_j) + m_{Fi}(\theta_k)} & \text{for } c_{Fi}(\theta_j) + c_{Fi}(\theta_k) \neq 0 \text{ and } m_{Fi}(\theta_j) \cdot m_{Fi}(\theta_k) \neq 0 \\
\frac{m_{Fi}(\theta_j \cap \theta_k)}{c_{Fi}(\theta_j) + c_{Fi}(\theta_k)} & \text{for } c_{Fi}(\theta_j) + c_{Fi}(\theta_k) \neq 0 \text{ and } m_{Fi}(\theta_j) \cdot m_{Fi}(\theta_k) = 0 \\
0 & \text{for } c_{Fi}(\theta_j) + c_{Fi}(\theta_k) = 0
\end{cases}
\tag{48}
$$

wherein

$$m_{Fi}(\theta_j \cap \theta_k) = m_F(\theta_k)m_i(\theta_j) + m_F(\theta_j)m_i(\theta_k) \tag{49}$$

$$m_{Fi}(\theta_j) = m_F(\theta_j) \cdot m_i(\theta_j) \tag{50}$$

$$c_{Fi}(\theta_j) = m_F(\theta_j) + m_i(\theta_j) \tag{51}$$

If at least one of the BBAs, $m_F(.)$ or $m_i(.)$, is zero, the fraction is discarded and the mass $m_{Fi}(\theta_j \cap \theta_k)$ is transferred to $\theta_j$ and $\theta_k$ proportionally, with respect to their non-zero column sum of masses $c_{Fi}(\theta_j)$.

### 3.2.6. The Proportional Conflict Redistribution Rule PCR5 for Two BBAs (Two Sources)

Similar to the PCR2-PCR4 rules, PCR5 redistributes the partial conflicting mass to the hypothesis involved in the partial conflict. PCR5 provides the most mathematically precise [6,18,22] redistribution of conflicting mass to non-empty sets in accordance with the logic of the conjunctive rule. However, it is more difficult to implement. The PCR5 rule is defined for every non-empty hypothesis $\theta_j \in \Theta$ in the following way:

$$
m'_F(\theta_j) = m_{PCR5}(\theta_j) = m_{Fi}(\theta_j) + \sum_{\substack{k=1,\ldots,6 \\ \theta_k \cap \theta_j = \varnothing}} S_{Fi}^{PCR5}(\theta_j, \theta_k) = m_{Fi}(\theta_j) + \sum_{\substack{k=1,\ldots,6 \\ k \neq j}} S_{Fi}^{PCR5}(\theta_j, \theta_k)
\tag{52}
$$

where

$$
S_{Fi}^{PCR5}(\theta_j, \theta_k) = 
\begin{cases}
\frac{m_F(\theta_j)^2 \cdot m_i(\theta_k)}{m_F(\theta_j) + m_i(\theta_k)} + \frac{m_i(\theta_j)^2 \cdot m_F(\theta_k)}{m_i(\theta_j) + m_F(\theta_k)} \text{ for} \\
m_F(\theta_j) + m_i(\theta_k) \neq 0 \text{ and } m_i(\theta_j) + m_F(\theta_k) \neq 0 \\
0 \text{ for } m_F(\theta_j) + m_i(\theta_k) = 0 \text{ or } m_i(\theta_j) + m_F(\theta_k) = 0
\end{cases}
\tag{53}
$$

wherein

$$m_{Fi}(\theta_j) = m_F(\theta_j) \cdot m_i(\theta_j) \tag{54}$$

In Formula (52), the component $S_{Fi}^{PCR5}$ is equal to zero if both denominators are equal to zero. In Formula (53), if a denominator is zero, then the component is discarded.

### 3.2.7. The Proportional Conflict Redistribution Rules PCR5 and PCR6 for Three BBAs (Three Sources)

In [6,22], improved proportional conflict redistribution rules of the combination of basic belief assignments PCR6, PCR5+, and PCR6+ are presented. The authors point out that these rules should be applied if, and only if, we are to combine more than two BBAs. If we only have two BBAs to combine (s = 2), we always obtain $m_{PCR5} = m_{PCR5+} = m_{PCR6} = m_{PCR6+}$, because in this case, the PCR5, PCR5+, PCR6, and PCR6+ rules coincide. Below are the formulas that define the PCR5 and PCR6 rules for three BBAs.

The PCR5 rule for three BBAs (three sources) is defined for every non-empty hypothesis in the following way:

$$m'_F(\theta_j) = m_{PCR5}(\theta_j) = \frac{m''(\theta_j)}{\sum_{i=1}^{6} m''(\theta_i)} \tag{55}$$

wherein

$$
\begin{aligned}
m''(\theta_j) &= m_{F12}(\theta_j) + \sum_{\substack{k=1,\dots,6 \\ l=1,\dots,6 \\ \theta_k \cap \theta_l \cap \theta_j = \varnothing}} S_{F12}^{PCR5}(\theta_j, \theta_k, \theta_l) + \sum_{\substack{k=1,\dots,6 \\ \theta_k \cap \theta_j = \varnothing}} S1_{F12}^{PCR5}(\theta_j, \theta_k) + + \sum_{\substack{k=1,\dots,6 \\ \theta_k \cap \theta_j = \varnothing}} S2_{F12}^{PCR5}(\theta_j, \theta_k) = \\
&= m_{F12}(\theta_j) + \sum_{\substack{k=1,\dots,6 \\ k \neq j}} \sum_{\substack{l=1,\dots,6 \\ l \neq j \wedge l \neq k}} S_{F12}^{PCR5}(\theta_j, \theta_k, \theta_l) + \sum_{\substack{k=1,\dots,6 \\ j \neq k}} S1_{F12}^{PCR5}(\theta_j, \theta_k) + + \sum_{\substack{k=1,\dots,6 \\ j \neq k}} S2_{F12}^{PCR5}(\theta_j, \theta_k) = \\
&= m_{F12}(\theta_j) + \sum_{\substack{k=1,\dots,6 \\ k \neq j}} \left[ \sum_{\substack{l=1,\dots,6 \\ l \neq j \wedge l \neq k}} S_{F12}^{PCR5}(\theta_j, \theta_k, \theta_l) + S1_{F12}^{PCR5}(\theta_j, \theta_k) + S2_{F12}^{PCR5}(\theta_j, \theta_k) \right]
\end{aligned} \tag{56}
$$

$$S_{F12}^{PCR5}(\theta_j, \theta_k, \theta_l) = \frac{m_F(\theta_j)^2 \cdot m_1(\theta_k) \cdot m_2(\theta_l)}{m_F(\theta_j) + m_1(\theta_k) + m_2(\theta_l)} + \frac{m_F(\theta_l) \cdot m_1(\theta_j)^2 \cdot m_2(\theta_k)}{m_F(\theta_l) + m_1(\theta_j) + m_2(\theta_k)} + \frac{m_F(\theta_k) \cdot m_1(\theta_l) \cdot m_2(\theta_j)^2}{m_F(\theta_k) + m_1(\theta_l) + m_2(\theta_j)} \tag{57}$$

$$S1_{F12}^{PCR5}(\theta_j, \theta_k) = \frac{m_F(\theta_j)^2 \cdot m_1(\theta_k) \cdot m_2(\theta_k)}{m_F(\theta_j) + m_1(\theta_k) + m_2(\theta_k)} + \frac{m_F(\theta_k) \cdot m_1(\theta_j)^2 \cdot m_2(\theta_k)}{m_F(\theta_k) + m_1(\theta_j) + m_2(\theta_k)} + \frac{m_F(\theta_k) \cdot m_1(\theta_k) \cdot m_2(\theta_j)^2}{m_F(\theta_k) + m_1(\theta_k) + m_2(\theta_j)} \tag{58}$$

$$S2_{F12}^{PCR5}(\theta_j, \theta_k) = \frac{m_F(\theta_j)^2 \cdot m_1(\theta_j)^2 \cdot m_2(\theta_k)}{m_F(\theta_j) + m_1(\theta_j) + m_2(\theta_k)} + \frac{m_F(\theta_k) \cdot m_1(\theta_j)^2 \cdot m_2(\theta_j)^2}{m_F(\theta_k) + m_1(\theta_j) + m_2(\theta_j)} + \frac{m_F(\theta_j)^2 \cdot m_1(\theta_k) \cdot m_2(\theta_j)^2}{m_F(\theta_j) + m_1(\theta_k) + m_2(\theta_j)} \tag{59}$$

$$m'_F(\theta_j) = m_{PCR5}(\theta_j) = \frac{m''(\theta_j)}{\sum_{i=1}^{6} m''(\theta_i)} \tag{60}$$

In Formulas (57)–(59), if a denominator is zero, then the component is discarded.

The quotient in Formula (55) ensures the normalization of the BBA vector $m'_F$, which ensures the following:

$$\sum_{i=1}^{6} m'_F(\theta_i) = \sum_{i=1}^{6} m_{PCR5}(\theta_i) = 1$$

The PCR6 rule for three BBAs (three sources) is defined for every non-empty hypothesis $\theta_j \in \Theta$ in the following way:

$$m'_F(\theta_j) = m_{PCR5}(\theta_j) = \frac{m''(\theta_j)}{\sum_{i=1}^{6} m''(\theta_i)} \tag{61}$$

wherein

$$
\begin{aligned}
m''(\theta_j) = {} & m_{F12}(\theta_j) + \sum_{\substack{k=1,\ldots,6 \\ l=1,\ldots,6 \\ \theta_k \cap \theta_l \cap \theta_j = \varnothing}} S^{PCR6}_{F12}(\theta_j, \theta_k, \theta_l) + \sum_{\substack{k=1,\ldots,6 \\ \theta_k \cap \theta_j = \varnothing}} S1^{PCR6}_{F12}(\theta_j, \theta_k) + \\
& + \sum_{\substack{k=1,\ldots,6 \\ \theta_k \cap \theta_j = \varnothing}} S2^{PCR6}_{F12}(\theta_j, \theta_k) = \\
= {} & m_{F12}(\theta_j) + \sum_{\substack{k=1,\ldots,6 \\ k\neq j}} \sum_{\substack{l=1,\ldots,6 \\ l\neq j \wedge l\neq k}} S^{PCR6}_{F12}(\theta_j, \theta_k, \theta_l) + \sum_{\substack{k=1,\ldots,6 \\ j\neq k}} S1^{PCR6}_{F12}(\theta_j, \theta_k) + \\
& + \sum_{\substack{k=1,\ldots,6 \\ j\neq k}} S2^{PCR6}_{F12}(\theta_j, \theta_k) = \\
= {} & m_{F12}(\theta_j) + \sum_{\substack{k=1,\ldots,6 \\ k\neq j}} \left[ \sum_{\substack{l=1,\ldots,6 \\ l\neq j \wedge l\neq k}} S^{PCR6}_{F12}(\theta_j, \theta_k, \theta_l) + S1^{PCR6}_{F12}(\theta_j, \theta_k) + S2^{PCR6}_{F12}(\theta_j, \theta_k) \right]
\end{aligned}
\tag{62}
$$

with

$$
S^{PCR5}_{F12}(\theta_j, \theta_k, \theta_l) = \frac{m_F(\theta_j)^2 \cdot m_1(\theta_k) \cdot m_2(\theta_l)}{m_F(\theta_j) + m_1(\theta_k) + m_2(\theta_l)} + \frac{m_F(\theta_l) \cdot m_1(\theta_j)^2 \cdot m_2(\theta_k)}{m_F(\theta_l) + m_1(\theta_j) + m_2(\theta_k)} + \frac{m_F(\theta_k) \cdot m_1(\theta_l) \cdot m_2(\theta_j)^2}{m_F(\theta_k) + m_1(\theta_l) + m_2(\theta_j)}
\tag{63}
$$

$$
S1^{PCR5}_{F12}(\theta_j, \theta_k) = \frac{m_F(\theta_j)^2 \cdot m_1(\theta_k) \cdot m_2(\theta_k)}{m_F(\theta_j) + m_1(\theta_k) + m_2(\theta_k)} + \frac{m_F(\theta_k) \cdot m_1(\theta_j)^2 \cdot m_2(\theta_k)}{m_F(\theta_k) + m_1(\theta_j) + m_2(\theta_k)} + \frac{m_F(\theta_k) \cdot m_1(\theta_k) \cdot m_2(\theta_j)^2}{m_F(\theta_k) + m_1(\theta_k) + m_2(\theta_j)}
\tag{64}
$$

$$
\begin{aligned}
S2^{PCR5}_{F12}(\theta_j, \theta_k) = {} & \frac{m_F(\theta_j)^2 \cdot m_1(\theta_j) \cdot m_2(\theta_k) + m_F(\theta_j) \cdot m_1(\theta_j)^2 \cdot m_2(\theta_k)}{m_F(\theta_j) + m_1(\theta_j) + m_2(\theta_k)} + \\
& + \frac{m_F(\theta_k) \cdot m_1(\theta_j)^2 \cdot m_2(\theta_j) + m_F(\theta_k) \cdot m_1(\theta_j) \cdot m_2(\theta_j)^2}{m_F(\theta_k) + m_1(\theta_j) + m_2(\theta_j)} + \\
& + \frac{m_F(\theta_j)^2 \cdot m_1(\theta_k) \cdot m_2(\theta_j) + m_F(\theta_j) \cdot m_1(\theta_k) \cdot m_2(\theta_j)^2}{m_F(\theta_j) + m_1(\theta_k) + m_2(\theta_j)}
\end{aligned}
\tag{65}
$$

and

$$
m_{F12}(\theta_j) = m_F(\theta_j) \cdot m_1(\theta_j) \cdot m_2(\theta_j)
\tag{66}
$$

In Formulas (63)–(65), if a denominator is zero, then the component is discarded.

The quotient in Formula (61) ensures the normalization of the BBA vector $m'_F$, which ensures the following:

$$
\sum_{i=1}^{6} m'_F(\theta_i) = \sum_{i=1}^{6} m_{PCR5}(\theta_i) = 1
$$

Comparing the two fusion schemes (Figures 2 and 3), it should be noted that sequential and global information fusion generally produces different results [18], i.e.,

$$
PCR5(m_F, m_1, m_2) \neq PCR5(PCR5(m_F, m_1), m_2) \neq PCR5(PCR5(m_F, m_2), m_1)
\tag{67}
$$

In addition, the article experimentally verified the theorem on the inequality of the results of both PCR5 and PCR6 rules for three BBAs (three sources) presented in [18]:

$$
PCR5(m_F, m_1, m_2) \neq PCR6(m_F, m_1, m_2)
\tag{68}
$$

## 4. Basic Belief Assignment for Combined Primary and Secondary Surveillance Radars

Combined primary and secondary (IFF) radars are the main source of identification information regarding air and maritime objects. A primary radar only yields the detection of an object in a supervised area. The detection of the object is the precondition for sending

a request to the object by the secondary radar (interrogator). Interpretation of the object response is dependent on the type of request. The so-called civilian modes only yield a determination of whether the detected object replies to an interrogation or not.

This paper assumes that the analyzed radar sensor consists of two radars: primary and secondary. Therefore, the probability of the correct detection and the correct identification of a target is expressed by the following formula:

$$P_{pi} = P_d \cdot P_{IFF} \tag{69}$$

where $P_d$ is the probability of correct detection of the target by a primary radar, and $P_{IFF}$ is the probability of a correct reply to an interrogation. If a target is detected by the primary radar and there is a lack of proper identification by the secondary radar, it can be assumed that the target has a value of attribute identification of UNKNOWN—U. Thus, the following relation can be written:

$$m(U) = P_d(1 - P_{IFF}), \tag{70}$$

where $m(U)$ is the mass of probability for a value of UNKNOWN identification attribute.

A method for calculating the probabilities $P_d$ and $P_{IFF}$ is presented in [7,17,23].

This section explains the way the remaining mass of probability is calculated $(1 - m(U))$. It is assumed there that every simulated target should have a base value of attribute identification from the set as follows:

$$\mathbf{Z}_{BI} = \{N_B, F_B, H_B\} \tag{71}$$

where:

- $N_B$: base NEUTRAL identity;
- $F_B$: base FRIEND identity;
- $H_B$: base HOSTILE identity.

STANAG 1241 introduces, in addition to the basic set of attribute identification values, secondary (additional) attribute identification values: SUSPECT (S) and ASSUMED FRIEND (AF). According to Figure 1, a table of possible attribute value transitions between set (10) and the set of secondary attribute identification values can be introduced:

$$\mathbf{Z}_{SI} = \{N_S, F_S, H_S, AF, S\} \tag{72}$$

The belief mass values contained in Table 1 determine how the mass of the base belief assignment is transformed into the mass of the secondary belief assignment. They can be estimated as empirical frequencies based on recorded archive events.

**Table 1.** Transformation of the base belief assignment mass into the secondary belief assignment mass.

| Base Identification $\rightarrow$ | $F_B$ | $N_B$ | $H_B$ |
|---|---|---|---|
| $F_S$ | $m(F_S\|F_B)$ | 0 | 0 |
| $N_S$ | 0 | $m(N_S\|N_B)$ | 0 |
| $H_S$ | 0 | 0 | $m(H_S\|H_B)$ |
| $AF$ | $m(AF\|F_B)$ | $m(AF\|N_B)$ | 0 |
| $S$ | 0 | $m(S\|F_B)$ | $m(S\|H_B)$ |

Of course, the normalization conditions are satisfied: $\sum_{x \in Z_{SI}} m(x|F_B) = 1$, $\sum_{x \in Z_{SI}} m(x|N_B) = 1$, and $\sum_{x \in Z_{SI}} m(x|H_B) = 1$.

The final values of the belief mass of secondary attribute identification values are calculated according to the formulas as follows:

1. For a target with the FRIEND base value of an attribute identification,

$$m(U) = P_d(1 - P_{IFF});$$
$$m(AF) = m(AF|F_B)(1 - m(U));$$
$$m(F_S) = m(F_S|F_B)(1 - m(U)).$$

2. For a target with the NEUTRAL base value of an attribute identification,

$$m(U) = P_d(1 - P_{IFF});$$
$$m(AF) = m(AF|N_B)(1 - m(U));$$
$$m(S) = m(S|N_B)(1 - m(U));$$
$$m(N_S) = (1 - m(AF|N_B) - m(S|N_B))(1 - m(U)).$$

3. For a target with the HOSTILE base value of an attribute identification,

$$m(U) = P_d(1 - P_{IFF});$$
$$m(H_S) = m(H_S|H_B)(1 - m(U));$$
$$m(S) = m(S|H_B)(1 - m(U)).$$

Other final values of the belief mass of secondary attribute identification values are equal to zero.

## 5. Basic Belief Assignment for ESM Sensors

ESM sensors consist of passive receivers and direction finders, which allow them to capture emitter signals coming from certain directions. In this way, the electronic recognition system can receive, among other data, information on radar emitters mounted on air or maritime platforms. Reports sent from the ESM sensors include, among others, the characteristics of the intercepted signal, the emitter's azimuth, and the so-called identification information.

This paper also assumes that sensors are equipped with specialized databases called the databases of emitter signal patterns, in which information about previously captured, processed, analyzed, recognized, and described radar emitter signals is stored, along with additional information about the type and mode of the emitter work, the platform on which these emitters can be installed, and the national or organizational affiliation of these platforms. The detected signals are the subject of an analysis procedure, which yields the determination of the so-called distinctive features of the signal, and then assigns this information to a specific electronic entity (already existing or created ad hoc) [24]. The basis for assigning distinctive information to an electronic entity is the azimuth angle of the incoming signal.

In the case of a high density of targets, identification information may fluctuate due to incorrect assignment of signal information to the electronic entity [25]. The impact of this negative phenomenon can be significantly reduced by an efficient estimation of the emitter positions [24]. Assuming that the sensors send all reports on the tracked electronic entities to the superior operation center in the electronic recognition system, such a center (in this paper, called the information-fusion center (IFC)) can perform the fusion function of the identification information. The fusion of identification information ensures greater stability of this information, i.e., resistance to accidental changes in sensor decisions.

An ESM sensor is a passive sensor that captures incoming electromagnetic signals generated firstly by radar emitters mounted on air or maritime platforms. This sensor recognizes radar signals determining the values of their distinctive features. In this paper, we do not handle methods of radar signals recognition in detail. We do, however, use the information about these methods to identify platforms generating the signals according to STANAG 1241—NATO Standardization Agreement and DSmT. As previously stated, we are interested in three basic values of identification: friend, hostile, and neutral, as well as two secondary values: suspicious and assumed friendly. In addition, we assume

that in some situations, it is not possible to determine the identity of the emitter-carrier platform. To clarify this issue, we briefly describe the method used to determine the identification of the emitter-carrier platform that generated the captured signal. The sensor-recognition system is equipped with a database that can be divided into three components: a platform database, an emitter list, and a geopolitical list [21]. The platform database (**PDB**) contains information about platforms that can be met in the area of interest, along with their equipment with emitters. The emitter name list (**ENL**) includes all emitters corresponding to each platform of the **PDB** and contains the values of the signal distinctive features for each emitter. The values of distinctive features are the basis for the procedure of recognizing a captured signal. The geopolitical list (**GPL**) provides the allegiance of various countries and platforms and yields the identification of them in accordance with STANAG 1241.

The algorithm of signal recognition is realized in two stages:

1. Verification at the level of signal quality features. The second stage is executed after a positive assessment of the conformity of quality features;

2. The signal-recognition procedure determines the distances between the distinctive features of the recognized signal and the distinctive features of all pattern signals stored within the emitter list.

Let us introduce the following notation:

$x_s$ : vector of distinctive features of the recognized signal;
$x_i$: vector of distinctive features of the *i*-th pattern signal (*i*: the number of the pattern signal, $i \in \{1, \ldots, M\}$);
$d_{s,i} = d(x_s, x_i)$ : the distance between the distinctive features vector of the recognized signal and the distinctive features vector of the i-th pattern signal; the distance $d_{s,i}$ is the Mahalanobis distance, taking into account the correlations of the distinctive features.

The signal-recognition classifier compares the distance $d(x_s, x_i)$ with the acceptable positive distance of the classification $\delta$. The distance $\delta$ is the limit that we interpret as a boundary of emitter pattern recognition. We divide the set of pattern signals into two subsets: the patterns satisfying the positive classification condition in relation to the recognized signal $s$ $\left(D_s^+\right)$ and the patterns that do not satisfy the positive classification condition $\left(D_s^-\right)$. The formal definition is as follows:

$$D_s^+ = \{\, i \in \{1, \ldots, M\} | d_{s,i} \leq \delta \} \tag{73}$$

$$D_s^- = \{\, i \in \{1, \ldots, M\} | d_{s,i} > \delta \} \tag{74}$$

In this paper, we propose the following method of determining the basic belief assignment on a set of pattern signals, which is related to the distance between a signal and a pattern in the distinctive features space:

$$m_s(i) = e^{-d(\mathbf{x}_s, \mathbf{x}_i)} \tag{75}$$

As can be seen from Formula (75), if $d(x_s, x_i) = 0$, then $m_s(i) = 1$, whereas if $d(x_s, x_i) > 0$, then $0 < m_s(i) < 1$. The above measure is not normalized; hence, we normalize it as follows:

$$\widetilde{m}_s(i) = \frac{m_s(i)}{\sum_{i=1}^{M} m_s(i)} \tag{76}$$

The sum of the measures assigned to all the emitters, the distinctive features of which lie outside the limit $\delta$, are treated as a measure assigned to the base hypothesis "unknown" (U):

$$\widetilde{m}_s(U) = \sum_{i \in D_s^-} \widetilde{m}_s(i) \tag{77}$$

Another way of recognizing emitters based on their signals is presented in [26]. For this, the authors use a convolutional neural network with a softmax layer.

To determine the belief measure of other base hypotheses (*H*, *F*, *N*) and secondary hypotheses (*AF* and *S*), we introduce formal definitions of sets contained in the sensor database and used for the recognition of captured signals. As mentioned above, the set of all the necessary data for platform identification can be divided into three sets: **PDB**, a platform database; **ENL**, an emitter name list; and **GPL**, a geopolitical list:

**PDB**: the platform database contains information about all platforms observed in the area of interest, including information on all emitters mounted on each platform; we assume that one platform can have many emitters and the same type of emitters can be installed on many platforms; the **PDB** also contains information on the national affiliation of each platform;

**ENL**: the emitter name list is a set of information about all recognized emitters in the area of interest; this set contains the mean values of the distinctive features of emitter signals (the so-called signal patterns) and their standard deviations;

**GPL**: the geopolitical list contains base values of identification attributes (*H*, *F*, *N*) assigned to the various countries.

We also introduce additional notations used in this paper:

- **PDBL**: the list of platform numbers that are stored in the **PDB**;
- **PL**(*i*): the set of numbers of platforms that have an emitter with number "*i*";

IPL(*j*): the base identification attribute of the platform with number "*j*" determined on the basis of the information contained in the **PDB** and **ENL** ($IPL(j) \in \{ F, H, N\}$).

The set of signal patterns satisfying the positive classification condition in relation to the recognized signal s, denoted as $\boldsymbol{D}_s^+$, can be divided into disjunctive subsets according to the values of the carrier platform identification features:

$$\boldsymbol{D}_s^+ = \boldsymbol{D}_s^{+F} \cup \boldsymbol{D}_s^{+H} \cup \boldsymbol{D}_s^{+N} \cup \boldsymbol{D}_s^{+AF} \cup \boldsymbol{D}_s^{+S}, \tag{78}$$

$$\boldsymbol{D}_s^{+k} \cap \boldsymbol{D}_s^{+l} = \varnothing, \; k \neq l, \; k,l \in \{F,H,N,AF,S\} \tag{79}$$

Each subset of the set $\boldsymbol{D}_s^+$ for the base identification is defined as follows:

$$\boldsymbol{D}_s^{+F} = \left\{ i \in \boldsymbol{D}_s^+ \,\middle|\, \forall j \in \boldsymbol{PL}(i) \;\; IPL(j) = F \right\}, \tag{80}$$

$$\boldsymbol{D}_s^{+H} = \left\{ i \in \boldsymbol{D}_s^+ \,\middle|\, \forall j \in \boldsymbol{PL}(i) \;\; IPL(j) = H \right\} \tag{81}$$

$$\boldsymbol{D}_s^{+N} = \left\{ i \in \boldsymbol{D}_s^+ \,\middle|\, \forall j \in \boldsymbol{PL}(i) \;\; IPL(j) = N \right\} \tag{82}$$

In a similar way, subsets of the set $\boldsymbol{D}_s^+$ for the secondary identification (*AF*, *S*) can be defined as follows:

$$\boldsymbol{D}_s^{+AF} = \left\{ i \in \boldsymbol{D}_s^+ \,\middle|\, \exists j \in \boldsymbol{PL}(i) \;\; IPL(j) = F \,\wedge\, \exists j \in \boldsymbol{PL}(i) \;\; IPL(j) = N \right\} \tag{83}$$

$$\mathbf{D}_s^{+S} = \left\{ i \in \boldsymbol{D}_s^+ : \exists j \in \boldsymbol{PL}(i) \;\; IPL(j) = H \,\wedge\, \exists j \in \boldsymbol{PL}(i) \;\; IPL(j) = N \right\} \tag{84}$$

It can be noticed that we assume in this paper that no emitter type can be installed simultaneously on platforms with identifications *F* and *H*:

$$\left\{ i \in \boldsymbol{D}_s^+ \,\middle|\, \exists j \in \boldsymbol{PL}(i) \;\; IPL(j) = F \;\; \wedge \;\; \exists j \in \boldsymbol{PL}(i) \;\; IPL(j) = H \right\} = \varnothing \tag{85}$$

Introducing the definition of subsets of the set determines the belief masses for all identification features:

$$\widetilde{m}_s(F) = \sum_{i \in \mathrm{D}_s^{+F}} \widetilde{m}_s(i), \; \widetilde{m}_s(H) = \sum_{i \in \mathrm{D}_s^{+H}} \widetilde{m}_s(i), \; \widetilde{m}_s(N) = \sum_{i \in \mathrm{D}_s^{+N}} \widetilde{m}_s(i), \tag{86}$$

$$\widetilde{m}_s(\mathrm{AF}) = \sum_{i \in \mathrm{D}_s^{+AF}} \widetilde{m}_s(i), \widetilde{m}_s(S) = \sum_{i \in \mathrm{D}_s^{+S}} \widetilde{m}_s(i) \tag{87}$$

It should be emphasized that the method presented here is different than that presented in [25,27]. These papers assume that ESM sensors can only generate basic declarations with attribute values FRIEND, HOSTILE, and NEUTRAL, but in this paper, we assume that ESM sensors can generate declarations from an extended set of attribute values (including ASSUME FRIEND, SUSPECT, and UNKNOWN).

## 6. Numerical Experiments of Fusion of Identification Information from ESM Sensors

*6.1. General Research Scheme of Fusion of Identification Information from ESM Sensors*

Figure 4 shows a general scheme of simulation experiments, which indicates the places of description of individual models.

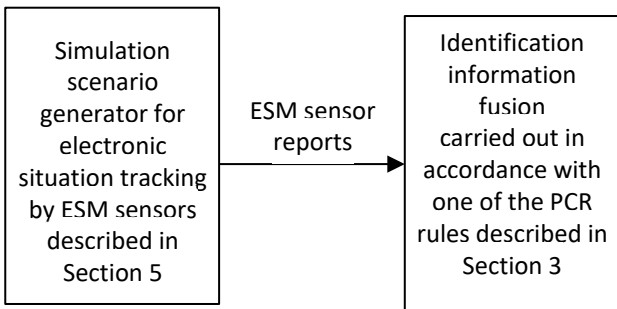

**Figure 4.** The general diagram of simulation experiments of fusion of identification information from ESM sensors.

*6.2. Simulation Scenarios*

Paper [25] presents a typical simulation scenario for testing identification information fusion. The authors formulated several requirements that should be met by such a scenario. It should meet the following requirements:

(1)  adequately represent the known ground truth of the emitter identification;
(2)  include sufficient numbers of incorrect associations to be realistic and to test the robustness of the rules in temporary incorrect sensor decisions;
(3)  provide only partial knowledge about the ESM sensor declarations and thus contain uncertainty;
(4)  be able to show stability in the case of countermeasures;
(5)  be able to switch identification when the ground truth changes.

The authors of [25] propose the following parameters of the scenario:

(1)  ground truth of identification is FRIEND (*F*) for the first 50 iterations of the scenario and HOSTILE (*H*) for the last 50 iterations;
(2)  the percentage of correct associations is 80% of all iterations, and the percentage of incorrect associations caused by countermeasures is 20% of all iterations in randomly selected moments of time;
(3)  ESM sensor declarations have a mass of 0.7 for the most credible identification and 0.3 for the identification of UNKNOWN (*U*).

Assumption (5) is not considered in this paper, assuming that the real object does not change its real identity while performing the mission. Therefore, assumption (1) regarding the scenario parameters becomes obsolete.

The following assumptions concerning the parameters of the scenario have been made in this paper:

(1)  the real value of identification is constant in each scenario and is equal to FRIEND (*F*) in scenarios 1, 2, and 5 and HOSTILE (*H*) in scenarios 3, 4, and 6;

(2)  the above declarations are transmitted by sensor number 1 with the real identification mass equal to 0.7 and the mass of complementary identification (UNKNOWN) equal to 0.3;

(3)  the second sensor transmits its declarations in accordance with Tables 2 and 3 for scenarios 1 and 2 and in accordance with Tables 4 and 5 for scenarios 3 and 4.

**Table 2.** Belief mass values for the second sensor for scenarios 1 and 5.

| Type of Identification | F | N | H | AF | S | U |
|---|---|---|---|---|---|---|
| Correct identification (80% of events) | 0.6 | 0.1 | 0 | 0.2 | 0 | 0.1 |
| Incorrect identification (20% of events) | 0 | 0.1 | 0.6 | 0 | 0.2 | 0.1 |

**Table 3.** Belief mass values for the second sensor for scenario 2.

| Type of Identification | F | N | H | AF | S | U |
|---|---|---|---|---|---|---|
| Correct identification (80% of events) | 0.7 | 0.1 | 0 | 0.1 | 0 | 0.1 |
| Incorrect identification (20% of events) | 0 | 0.1 | 0.7 | 0 | 0.7 | 0.1 |

**Table 4.** Belief mass values for the second sensor for scenario 3.

| Type of Identification | F | N | H | AF | S | U |
|---|---|---|---|---|---|---|
| Correct identification (80% of events) | 0 | 0.1 | 0.6 | 0 | 0.2 | 0.1 |
| Incorrect identification (20% of events) | 0.6 | 0.1 | 0 | 0.2 | 0 | 0.1 |

**Table 5.** Belief mass values for the second sensor for scenarios 4 and 6.

| Type of Identification | F | N | H | AF | S | U |
|---|---|---|---|---|---|---|
| Correct identification (80% of events) | 0 | 0.1 | 0.7 | 0 | 0.1 | 0.1 |
| Incorrect identification (20% of events) | 0.7 | 0.1 | 0 | 0.1 | 0 | 0.1 |

It should be noted that scenario 2 differs from scenario 1 by a greater belief mass assigned to an incorrect identification of the recognized emitter. Scenarios 3 and 4 are similarly different.

Scenarios 1–6 for sensor 1 are presented in Figures 5 and 6. All deterministic scenarios for sensor 2 are presented in Figures 7–10.

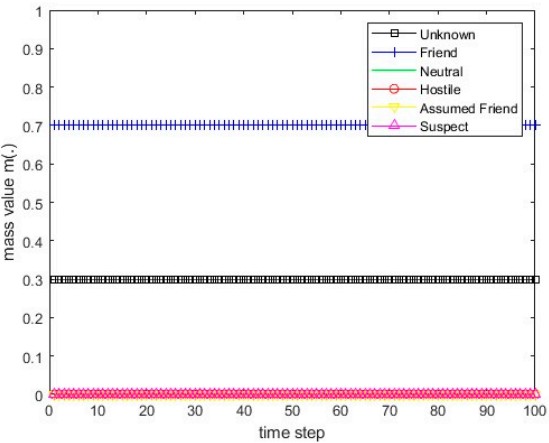

**Figure 5.** The course of scenarios number 1, 2, and 5 for sensor 1.

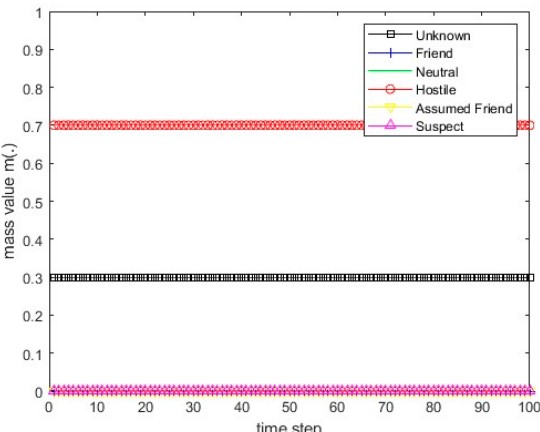

**Figure 6.** The course of scenarios number 3, 4, and 6 for sensor 1.

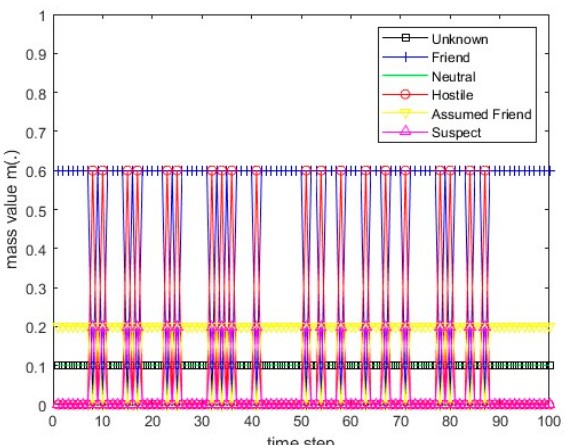

**Figure 7.** The course of scenario number 1 for sensor 2.

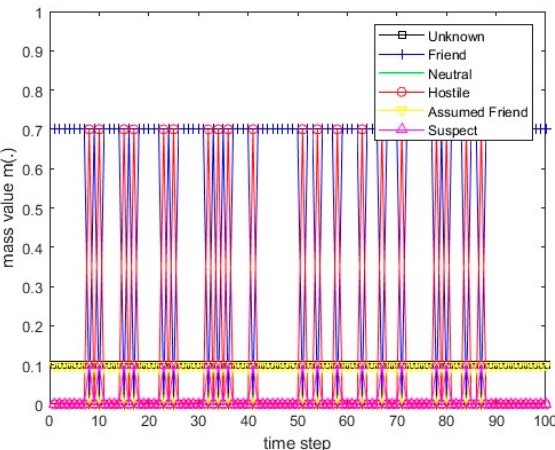

**Figure 8.** The course of scenario number 2 for sensor 2.

This section is divided by subheadings. It provides a concise and precise description of the experimental results, their interpretation, and the experimental conclusions that can be drawn.

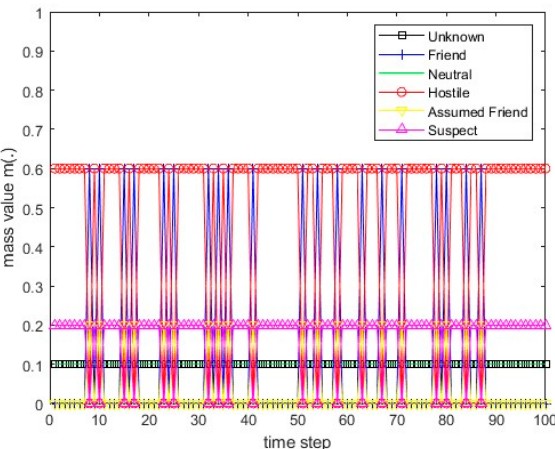

**Figure 9.** The course of scenario number 3 for sensor 2.

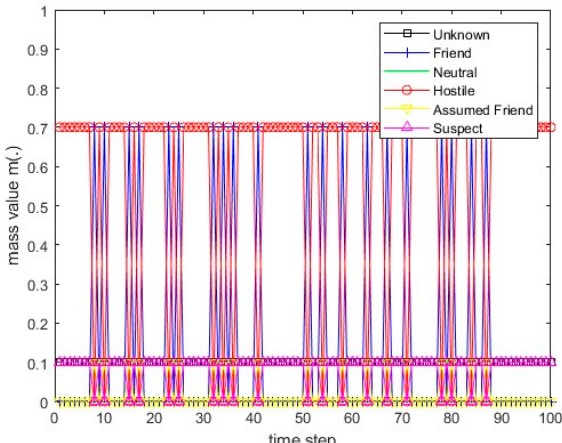

**Figure 10.** The course of scenario number 4 for sensor 2.

The paper also uses the Monte Carlo method for generating the scenario for sensor 2. Moments in which incorrect identifications occurred are generated by the pseudorandom integer number generator from the range [0, 100]. Examples of scenarios are shown in Figures 11 and 12.

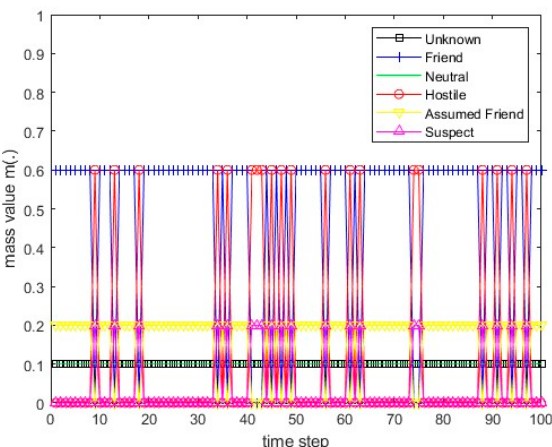

**Figure 11.** The course of Monte Carlo scenario number 5 for sensor 2.

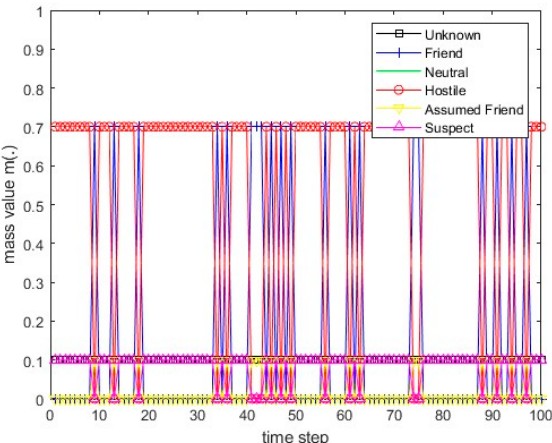

**Figure 12.** The course of Monte Carlo scenario number 6 for sensor 2.

### 6.3. Calculation Results for Deterministic Scenarios

In all figures presenting the values of the resulting belief mass, the decision threshold is marked with a horizontal line. An identification whose belief mass at a given moment is above the decision threshold is the so-called hard decision.

#### 6.3.1. Dempster's Rule

Dempster's rule is not resistant to a situation where the degree of conflict $k_{Fi} = 1$. This means the total conflict between the mass vector sent by the sensor and the mass vector of the information-fusion center, which occurs when each non-zero belief mass value sent by the sensor corresponds to the zero belief mass value of the vector determined by the information-fusion center and vice versa.

The simulation results of identification information fusion using Dempster's rule are presented for deterministic scenarios 1 and 3 in Figures 13 and 14, respectively. When the degree of conflict $k_{Fi} = 1$, according to Equation (8), it is impossible to perform sensor information fusion, i.e., it is impossible to determine the resulting BBA.

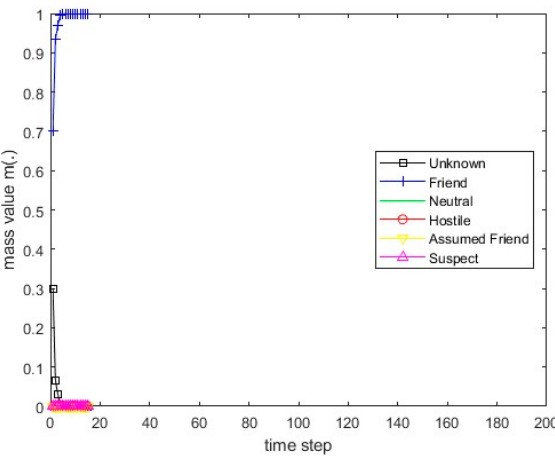

**Figure 13.** The values of the resulting belief mass for scenario 1 and Dempster's rule.

#### 6.3.2. The PCR1 Rule

The simulation results of identification information fusion using the PCR1 rule for deterministic scenarios 1, 2, 3, and 4 are presented in Figures 15–18, respectively.

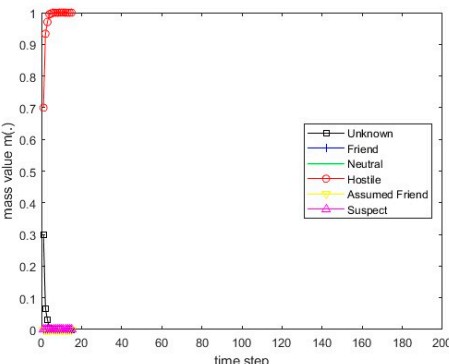

**Figure 14.** The values of the resulting belief mass for scenario 3 and Dempster's rule.

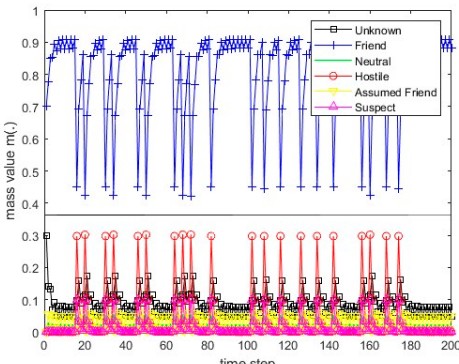

**Figure 15.** The values of the resulting belief mass for scenario 1 and the PCR1 rule.

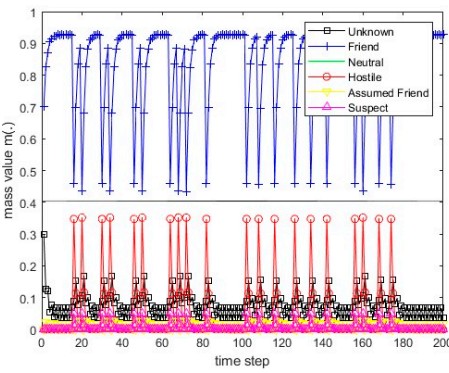

**Figure 16.** The values of the resulting belief mass for scenario 2 and the PCR1 rule.

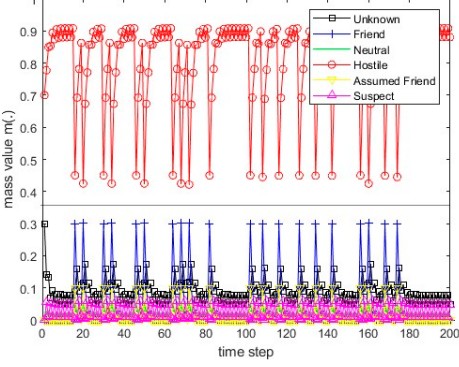

**Figure 17.** The values of the resulting belief mass for scenario 3 and the PCR1 rule.

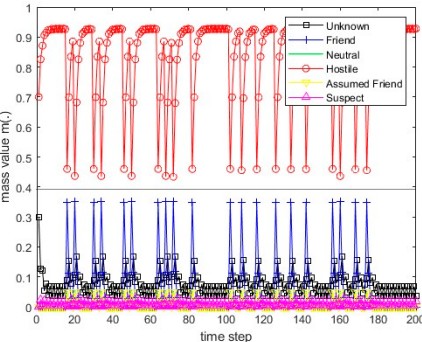

**Figure 18.** The values of the resulting belief mass for scenario 4 and the PCR1 rule.

### 6.3.3. The PCR3 Rule

The simulation results of identification information fusion using the PCR3 rule for deterministic scenarios 1, 2, 3, and 4 are presented in Figures 19–22, respectively.

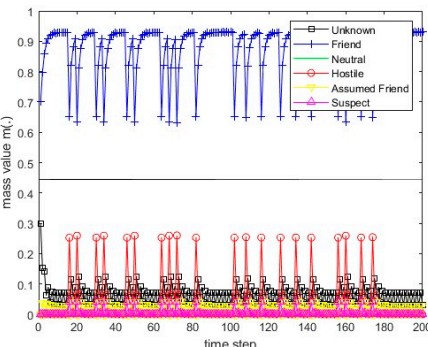

**Figure 19.** The values of the resulting belief mass for scenario 1 and the PCR3 rule.

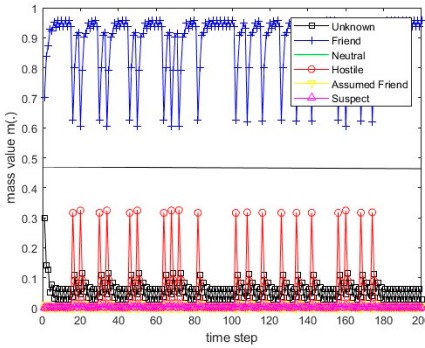

**Figure 20.** The values of the resulting belief mass for scenario 2 and the PCR3 rule.

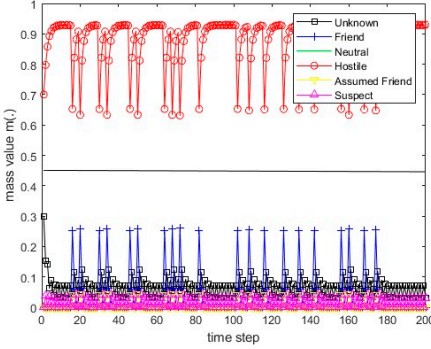

**Figure 21.** The values of the resulting belief mass for scenario 3 and the PCR3 rule.

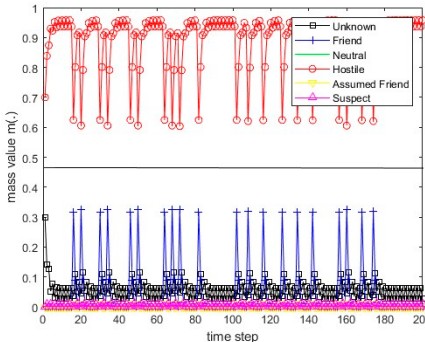

**Figure 22.** The values of the resulting belief mass for scenario 4 and the PCR3 rule.

### 6.3.4. The PCR4 Rule

The simulation results of identification information fusion using the PCR4 rule for deterministic scenarios 1, 2, 3, and 4 are presented in Figures 23–26, respectively.

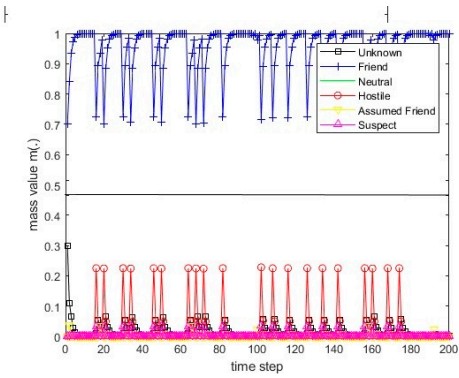

**Figure 23.** The values of the resulting belief mass for scenario 1 and the PCR4 rule.

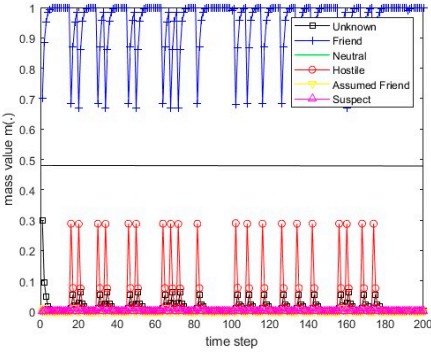

**Figure 24.** The values of the resulting belief mass for scenario 2 and the PCR4 rule.

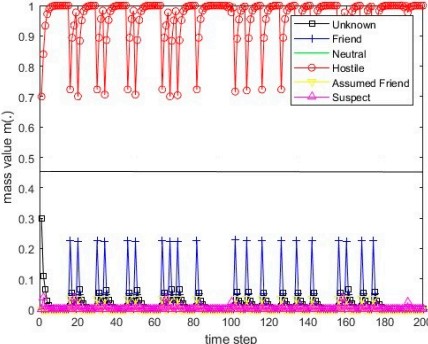

**Figure 25.** The values of the resulting belief mass for scenario 3 and the PCR4 rule.

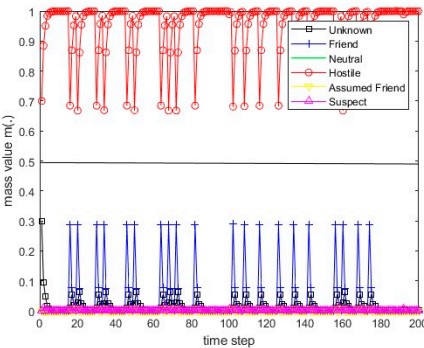

**Figure 26.** The values of the resulting belief mass for scenario 4 and the PCR4 rule.

### 6.3.5. The PCR5 Rule for 2 BBAs

The simulation results of identification information fusion using the PCR5 rule for two BBAs for deterministic scenarios 1, 2, 3, and 4 are presented in Figures 27–30, respectively.

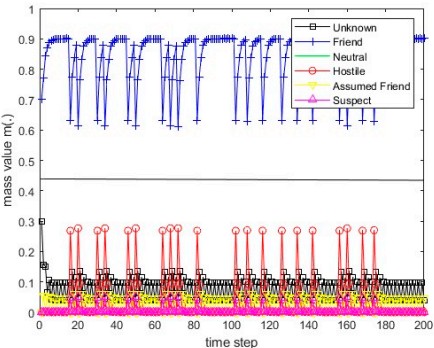

**Figure 27.** The values of the resulting belief mass for scenario 1 and the PCR5 rule for 2 BBAs.

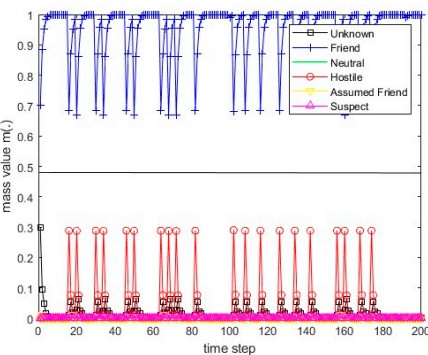

**Figure 28.** The values of the resulting belief mass for scenario 2 and the PCR5 rule for 2 BBAs.

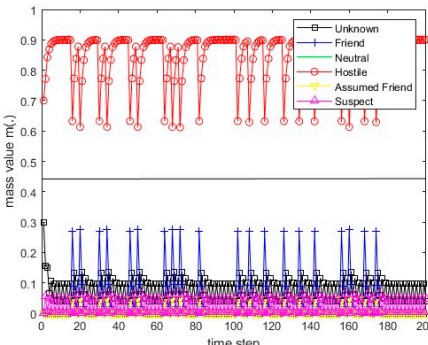

**Figure 29.** The values of the resulting belief mass for scenario 3 and the PCR5 rule for 2 BBAs.

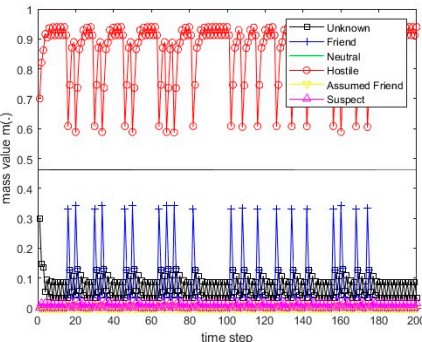

**Figure 30.** The values of the resulting belief mass for scenario 4 and the PCR5 rule for 2 BBAs.

### 6.3.6. The PCR5 Rule for 3 BBAs

The simulation results of identification information fusion using the PCR5 rule for three BBAs for deterministic scenarios 1, 2, 3, and 4 are presented in Figures 31–34, respectively.

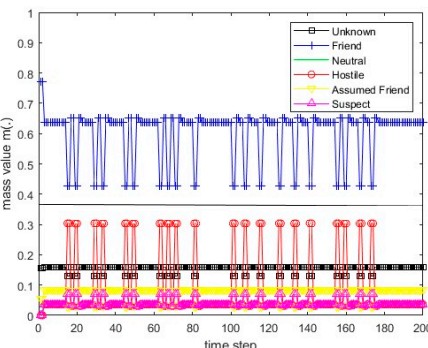

**Figure 31.** The values of the resulting belief mass for scenario 1 and the PCR5 rule for 3 BBAs.

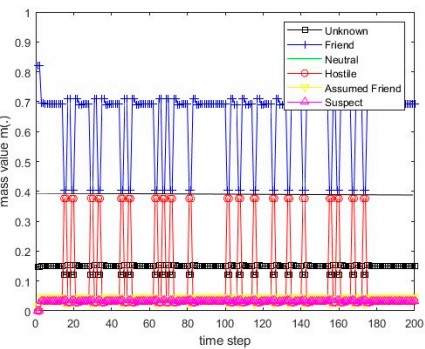

**Figure 32.** The values of the resulting belief mass for scenario 2 and the PCR5 rule for 3 BBAs.

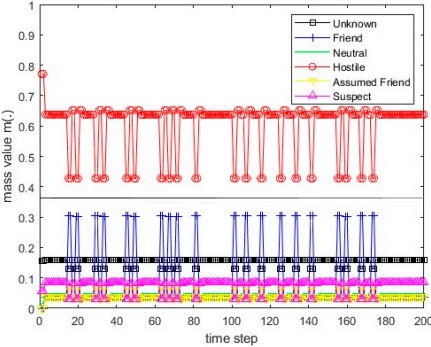

**Figure 33.** The values of the resulting belief mass for scenario 3 and the PCR5 rule for 3 BBAs.

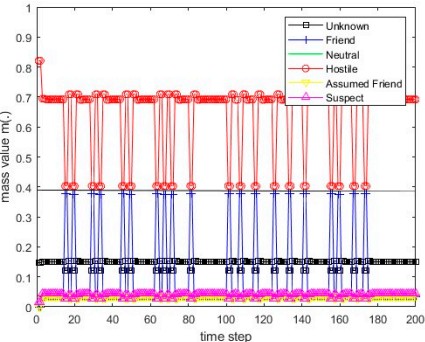

**Figure 34.** The values of the resulting belief mass for scenario 4 and the PCR5 rule for 3 BBAs.

### 6.3.7. The PCR6 Rule for 3 BBAs

The simulation results of identification information fusion using the PCR6 rule for three BBAs for deterministic scenarios 1, 2, 3, and 4 are presented in Figures 35–38, respectively.

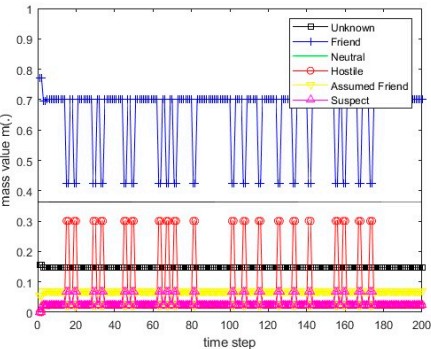

**Figure 35.** The values of the resulting belief mass for scenario 1 and the PCR6 rule for 3 BBAs.

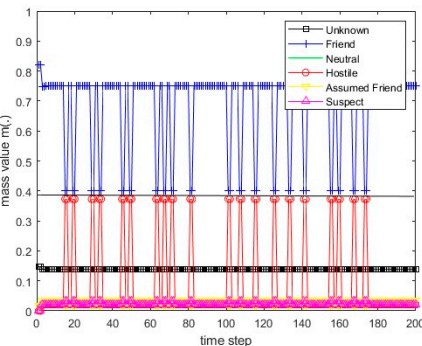

**Figure 36.** The values of the resulting belief mass for scenario 2 and the PCR6 rule for 3 BBAs.

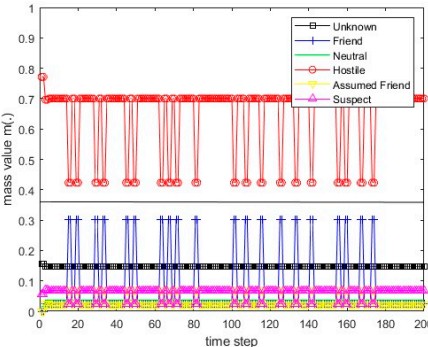

**Figure 37.** The values of the resulting belief mass for scenario 3 and the PCR6 rule for 3 BBAs.

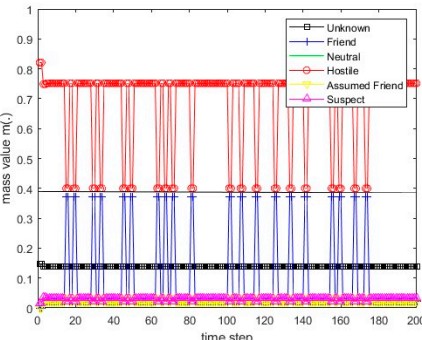

**Figure 38.** The values of the resulting belief mass for scenario 4 and the PCR6 rule for 3 BBAs.

The presented results (Figures 13–38) yield the conclusion that the applied methods of managing conflicts in information fusion enables correct conclusions to be drawn about the real identification of the recognized object.

The application of the decision threshold for the belief mass at the level $m_\alpha = 0.37$ for the PCR1 rule (Figures 15 and 17) and $m_\alpha = 0.45$ for PCR3, PCR4 (Figures 19, 21, 23 and 25), and PCR5 for two BBAs (Figures 27 and 29) for scenarios 1 and 3 allows for a proper evaluation of the identification of the recognized object: scenario 1, FRIEND; scenario 3, HOSTILE. For scenarios 2 and 4, the optimal thresholds are $m_\alpha = 0.4$ for the PCR1 rule (Figures 16 and 18) and $m_\alpha = 0.48$ for the PCR3, PCR4 (Figures 20, 22, 24 and 26), and PCR5 rules for two BBAs (Figures 28 and 30). When assessing the interval between the minimum resultant mass for correct identification and the maximum resultant mass for misidentification, the worst results are reached by the PCR1 rule, and the rules of PCR3, PCR4, and PCR5 behave similarly and are better than rule PCR1.

The research carried out for the deterministic scenarios shows that the PCR5 rule for three BBAs and the PCR6 rule for three BBAs behave very similarly (Figures 31, 33, 35 and 37 for scenarios 1 and 3, $m_\alpha = 0.37$ and Figures 32, 34, 36 and 37 for scenarios 2 and 4, $m_\alpha = 0.39$). They restore the correct identification after the occurrence of temporary misidentification much faster than the rules PCR1–PCR5 for two BBAs.

### 6.4. Calculation Results for the Monte Carlo Scenarios

#### 6.4.1. Dempster's Rule

In the Monte Carlo scenario, Dempster's rule behaves similarly to a deterministic scenario. It is not resistant to a situation where the degree of conflict $k_{Fi} = 1$. This means that the total conflict between the mass vector sent by the sensor and the mass vector of the information-fusion center, which occurs when each non-zero belief mass value sent by the sensor corresponds to the zero belief mass value of the vector determined by the information-fusion center and vice versa.

The simulation results of identification information fusion using Dempster's rule are presented for Monte Carlo scenarios 5 and 6 in Figures 39 and 40, respectively.

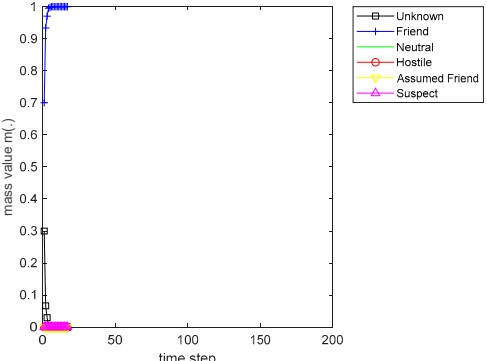

**Figure 39.** The values of the resulting belief mass for Monte Carlo scenario 5 and Dempster's rule.

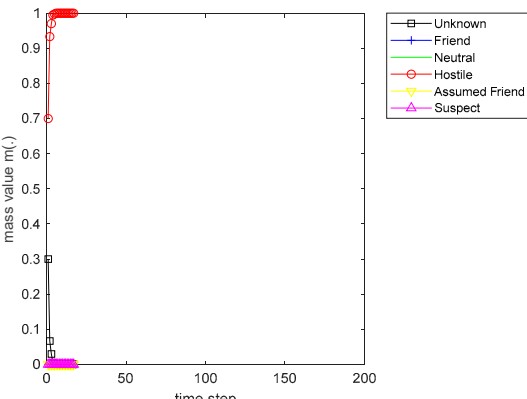

**Figure 40.** The values of the resulting belief mass for Monte Carlo scenario 6 and Dempster's rule.

### 6.4.2. The PCR1 Rule

The simulation results of identification information fusion using the PCR1 rule for Monte Carlo scenarios 5 and 6 are presented in Figures 41 and 42, respectively. The application of the decision threshold for the belief mass at the level $m_\alpha = 0.34$ for the PCR1 rule (Figures 41 and 42) for scenarios 5 and 6 allows for a proper evaluation of the identification of the recognized object. There are only three time points when the rule misidentifies due to increased misidentification intensity.

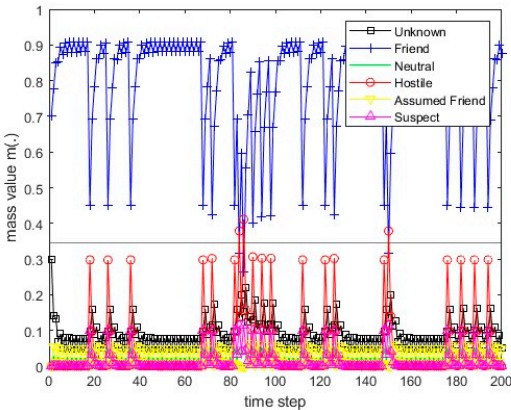

**Figure 41.** The values of the resulting belief mass for Monte Carlo scenario 5 and the PCR1 rule.

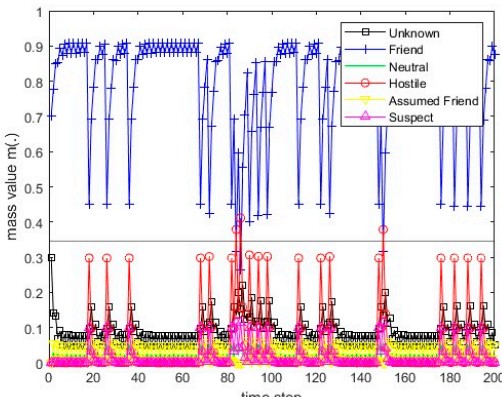

**Figure 42.** The values of the resulting belief mass for Monte Carlo scenario 6 and the PCR1 rule.

### 6.4.3. The PCR3 Rule

The simulation results of identification information fusion using the PCR3 rule for Monte Carlo scenarios 5 and 6 are presented in Figures 43 and 44, respectively. The

application of the decision threshold for the belief mass at the level $m_\alpha = 0.42$ for the PCR3 rule (Figures 43 and 44) for scenarios 5 and 6 allows for a proper evaluation of the identification of the recognized object. There is only one time point for scenario 6 where the rule misidentifies due to increased misidentification intensity.

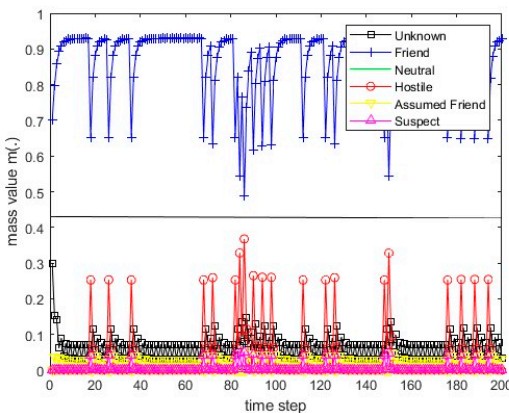

**Figure 43.** The values of the resulting belief mass for Monte Carlo scenario 5 and the PCR3 rule.

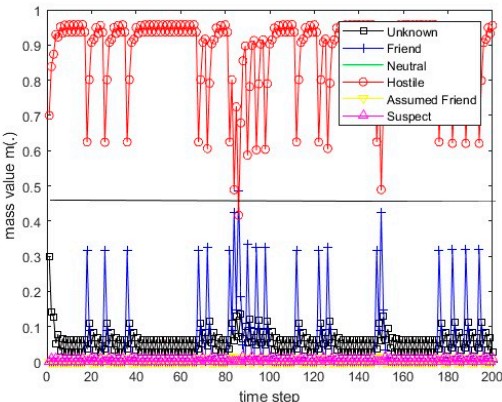

**Figure 44.** The values of the resulting belief mass for Monte Carlo scenario 6 and the PCR3 rule.

6.4.4. The PCR4 Rule

The simulation results of identification information fusion using the PCR4 rule for Monte Carlo scenarios 5 and 6 are presented in Figures 45 and 46, respectively. The application of the decision threshold for the belief mass at the level $m_\alpha = 0.47$ for the PCR4 rule (Figures 45 and 46) for scenarios 5 and 6 allows for a proper evaluation of the identification of the recognized object.

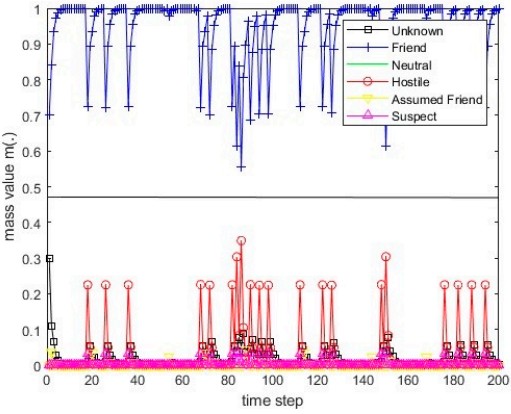

**Figure 45.** The values of the resulting belief mass for Monte Carlo scenario 5 and the PCR4 rule.

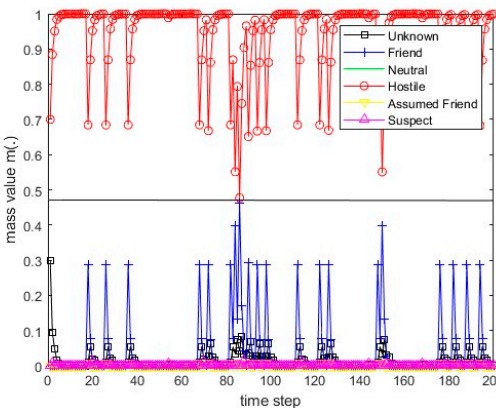

**Figure 46.** The values of the resulting belief mass for Monte Carlo scenario 6 and the PCR4 rule.

### 6.4.5. The PCR5 Rule for 2 BBAs

The simulation results of identification information fusion using the PCR5 rule for two BBAs for Monte Carlo scenarios 5 and 6 are presented in Figures 47 and 48, respectively. The application of the decision threshold for the belief mass at the level $m_\alpha = 0.47$ for the PCR5 rule for two BBAs (Figures 47 and 48) for scenarios 5 and 6 allows for a proper evaluation of the identification of the recognized object. There is only one time point for scenario 6 where the rule misidentifies due to increased misidentification intensity.

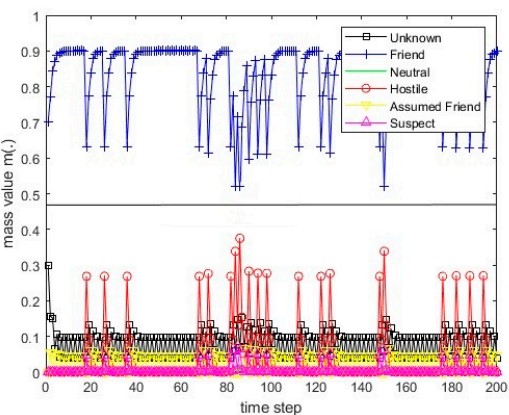

**Figure 47.** The values of the resulting belief mass for Monte Carlo scenario 5 and the PCR5 rule for 2 BBAs.

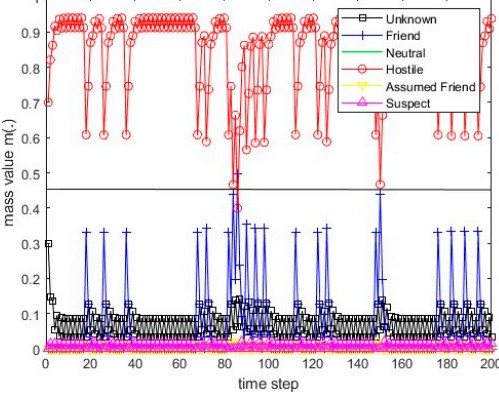

**Figure 48.** The values of the resulting belief mass for Monte Carlo scenario 6 and the PCR5 rule for 2 BBAs.

### 6.4.6. The PCR5 Rule for 3 BBAs

The simulation results of identification information fusion using the PCR5 rule for three BBAs for Monte Carlo scenarios 5 and 6 are presented in Figures 49 and 50, respectively. The application of the decision threshold for the belief mass at the level $m_\alpha = 0.39$ for the PCR5 rule for three BBAs (Figures 49 and 50) for scenarios 5 and 6 allows for a proper evaluation of the identification of the recognized object.

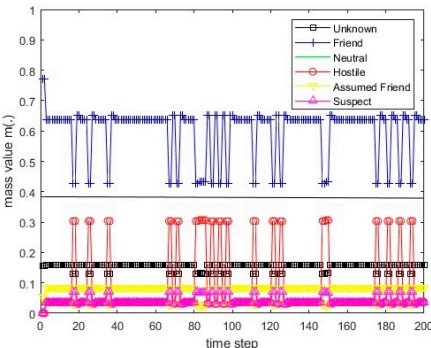

**Figure 49.** The values of the resulting belief mass for Monte Carlo scenario 5 and the PCR5 rule for 3 BBAs.

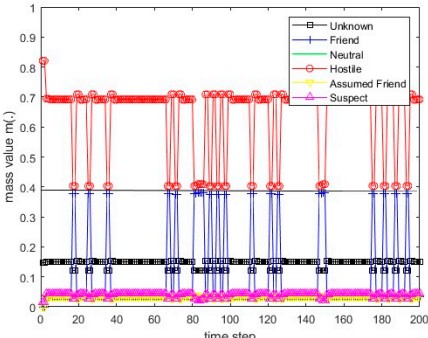

**Figure 50.** The values of the resulting belief mass for Monte Carlo scenario 6 and the PCR5 rule for 3 BBAs.

### 6.4.7. The PCR6 Rule for 3 BBAs

The simulation results of identification information fusion using the PCR6 rule for three BBAs for Monte Carlo scenarios 5 and 6 are presented in Figures 51 and 52. The application of the decision threshold for the belief mass at the level $m_\alpha = 0.39$ for the PCR6 rule for three BBAs (Figures 51 and 52) for scenarios 5 and 6 allows for a proper evaluation of the identification of the recognized object.

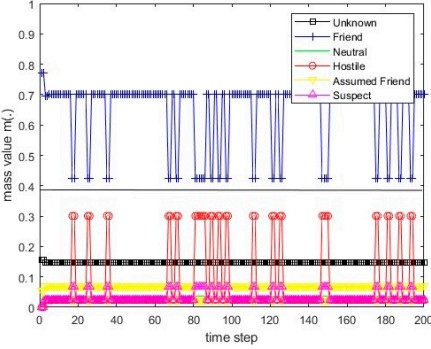

**Figure 51.** The values of the resulting belief mass for Monte Carlo scenario 5 and the PCR6 rule for 3 BBAs.

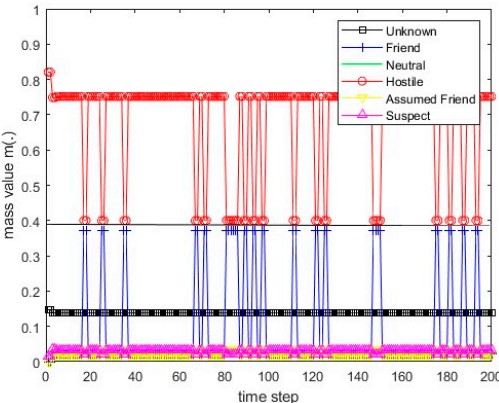

**Figure 52.** The values of the resulting belief mass for Monte Carlo scenario 6 and the PCR6 rule for 3 BBAs.

The presented results show that due to the high intensity of sending reports with incorrect identifications in the middle part of the scenarios, the information-fusion rules (apart from the PCR4, PCR5, and PCR6 rules) determine the maximum resulting mass for incorrect identification. The PCR5 for three BBAs and PCR6 for three BBAs rules are the fastest to restore the correct identification after receiving several incorrect reports.

## 7. Numerical Experiments of Fusion of Identification Information from ESM Sensors and Radars

### 7.1. General Research Scheme of Fusion of Identification Information from Radars and ESM Sensors

Figure 53 shows a general scheme of simulation experiments, which indicates the places of description of individual models.

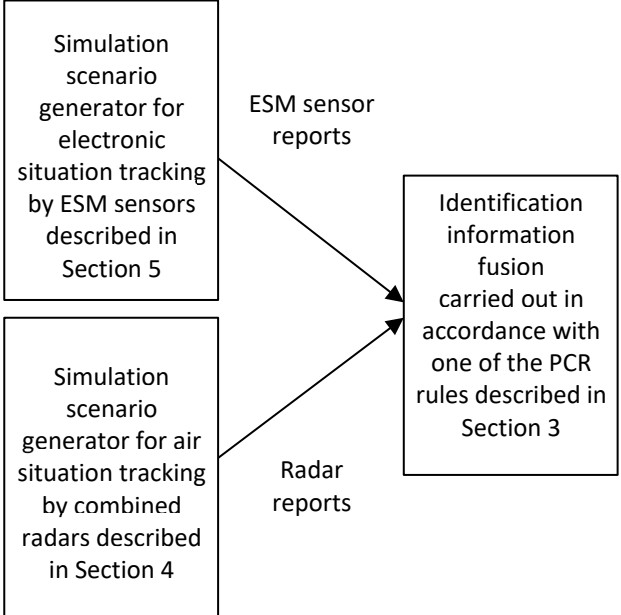

**Figure 53.** The general diagram of simulation experiments of fusion of identification information from radars and ESM sensors.

### 7.2. Numerical Experiments Scenarios

We assume that we will combine attribute information from two sensors: a combined primary and secondary surveillance radar and ESM sensor. These sensors work asynchronously. Upon receipt of the sensor's declaration in the form of a vector of masses, we

fuse this vector with the vector of the actual values of the declaration masses for the fuser's frame of discernment. The frequency of transmission of the sensor declarations depends on the rules of the data exchange network and on the technical characteristics of the sensors. Various combination methods are presented in [3,4]. This paper used two of the methods of proportional redistribution conflict (PRC5 and PCR6 [6,22]). Information fusion has been simulated for two processing schemes (Figures 2 and 3). The numerical model of combined primary and secondary surveillance radars was taken from [17,23]. It allows for the determination of the probability $P_d$ during the simulation of the object's movement, i.e., the change in the object's position relative to the radar. Detailed rules for determining BBAs' vectors, assuming the knowledge of probabilities $P_d$ and $P_{IFF}$, are presented in Section 4.

Numerical experiments have been performed for the following data:

- for combined primary and secondary surveillance radars sensor:

$$P_{fa} = 10^{-6}, \; R^*_{max} = 100 \; [\text{km}], \; P^*_d = 0.7, \; \sigma^*_c = 2 \; \left[\text{m}^2\right], \; P_{IFF} = 0.962$$

and the following table of masses (compare Table 6):

**Table 6.** Transformation of the base belief assignment mass into the secondary belief assignment mass for combined primary and secondary surveillance radar.

| (Scenario Nr, Base Identification) → | (1,$F_B$) | (2,$N_B$) | (3,$H_B$) |
|:---:|:---:|:---:|:---:|
| $F_S$ | 0.8 | 0 | 0 |
| $N_S$ | 0 | 0.5 | 0 |
| $H_S$ | 0 | 0 | 0.7 |
| $AF$ | 0.2 | 0.3 | 0 |
| $S$ | 0 | 0.2 | 0.3 |

The flight path of the air object was 30 km away from the sensor (in the horizontal plane), the flight altitude was 1 km, and the radar cross-section was 1 m$^2$.

The following assumptions concerning the parameters of the scenario for the ESM sensor were made in this paper:

(1) The real value of identification is constant in each scenario and is equal to FRIEND (F) in the first scenario and HOSTILE (H) in the second scenario;

(2) The above declarations are transmitted by sensor number 1 with the real identification mass equal to 0.7 and the mass of complementary identification (UNKNOWN) equal to 0.3;

(3) The second sensor shall transmit its declarations in accordance with Tables 1 and 2 for scenarios 1 and 2, respectively, and with Tables 3 and 4 for scenarios 3 and 4, respectively.

Tables 7–12 present the mass values for all possible declarations for the six scenarios for the ESM sensor.

**Table 7.** Belief mass values for the second sensor (ESM) for scenario 1.

| Type of Identification | F | N | H | AF | S | U |
|:---|:---:|:---:|:---:|:---:|:---:|:---:|
| Correct identification (80% of events) | 0.6 | 0.1 | 0 | 0.2 | 0 | 0.1 |
| Incorrect identification (20% of events) | 0 | 0.1 | 0.6 | 0 | 0.2 | 0.1 |

**Table 8.** Belief mass values for the second sensor (ESM) for scenario 2.

| Type of Identification | F | N | H | AF | S | U |
|:---|:---:|:---:|:---:|:---:|:---:|:---:|
| Correct identification (80% of events) | 0 | 0.5 | 0.3 | 0 | 0.2 | 0 |
| Incorrect identification (20% of events) | 0 | 0.4 | 0.2 | 0 | 0.3 | 0.1 |

**Table 9.** Belief mass values for the second sensor (ESM) for scenario 3.

| Type of Identification | *F* | *N* | *H* | *AF* | *S* | *U* |
| --- | --- | --- | --- | --- | --- | --- |
| Correct identification (80% of events) | 0 | 0.1 | 0.7 | 0 | 0.1 | 0.1 |
| Incorrect identification (20% of events) | 0 | 0.1 | 0.6 | 0 | 0.2 | 0.1 |

**Table 10.** Belief mass values for the second sensor (ESM) for scenario 4.

| Type of Identification | *F* | *N* | *H* | *AF* | *S* | *U* |
| --- | --- | --- | --- | --- | --- | --- |
| Correct identification (80% of events) | 0.1 | 0.7 | 0.1 | 0 | 0 | 0.1 |
| Incorrect identification (20% of events) | 0 | 0.1 | 0.6 | 0 | 0.2 | 0.1 |

**Table 11.** Belief mass values for the second sensor (ESM) for scenario 5.

| Type of Identification | *F* | *N* | *H* | *AF* | *S* | *U* |
| --- | --- | --- | --- | --- | --- | --- |
| Correct identification (80% of events) | 0.6 | 0.1 | 0 | 0.2 | 0 | 0.1 |
| Incorrect identification (20% of events) | 0 | 0.1 | 0.6 | 0 | 0.2 | 0.1 |

**Table 12.** Belief mass values for the second sensor (ESM) for scenario 6.

| Type of Identification | *F* | *N* | *H* | *AF* | *S* | *U* |
| --- | --- | --- | --- | --- | --- | --- |
| Correct identification (80% of events) | 0.1 | 0.7 | 0.1 | 0 | 0 | 0.1 |
| Incorrect identification (20% of events) | 0.6 | 0.1 | 0 | 0.2 | 0 | 0.1 |

Scenarios 1–6 for sensor 1 are presented in Figures 54–56. All deterministic scenarios for sensor 2 are presented in Figures 57–62.

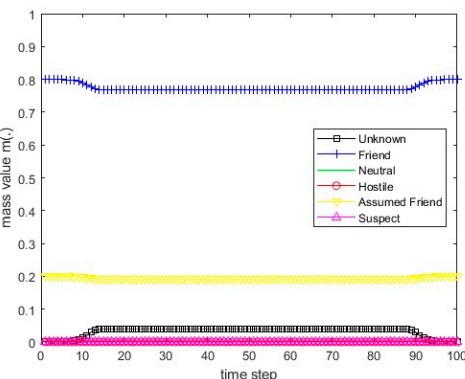

**Figure 54.** The course of scenarios number 1 and 4 for sensor 1.

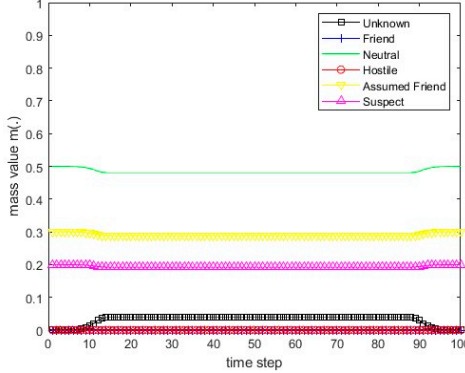

**Figure 55.** The course of scenarios number 2 and 5 for sensor 1.

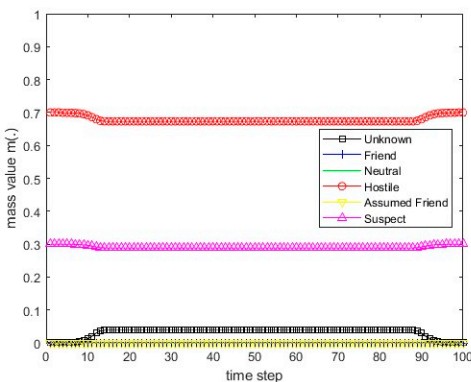

**Figure 56.** The course of scenarios number 3 and 6 for sensor 1.

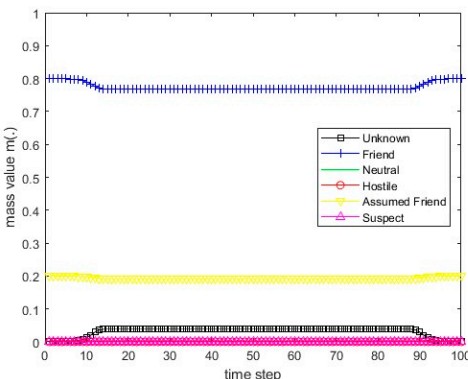

**Figure 57.** The course of scenario number 1 for sensor 2.

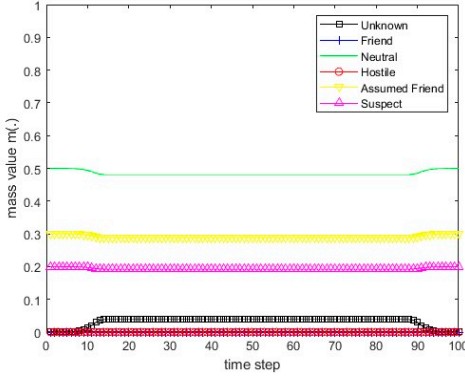

**Figure 58.** The course of scenario number 2 for sensor 2.

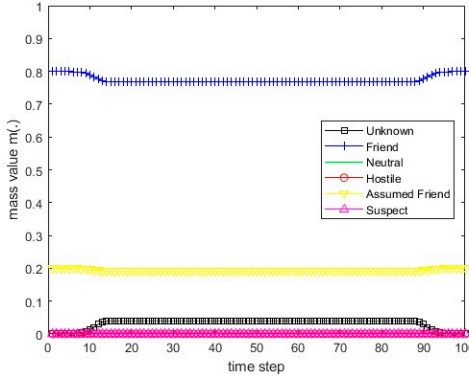

**Figure 59.** The course of scenario number 3 for sensor 2.

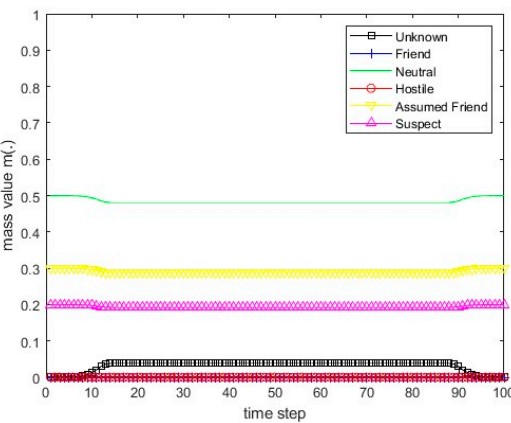

**Figure 60.** The course of scenario number 4 for sensor 2.

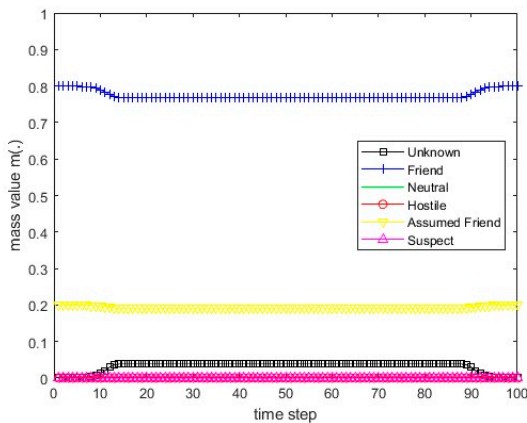

**Figure 61.** The course of scenario number 5 for sensor 2.

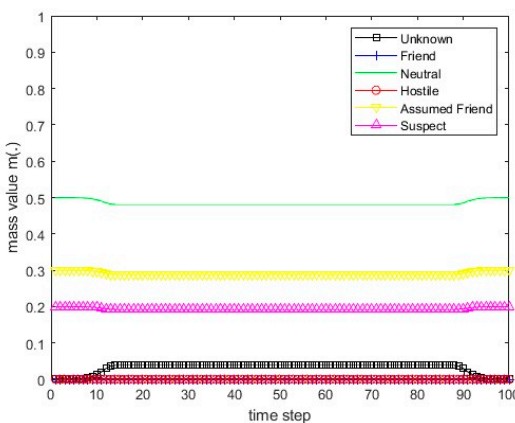

**Figure 62.** The course of scenario number 6 for sensor 2.

Scenarios 1–3 assume relatively small changes in the mass of all declarations. Scenarios 1–3 assume significant changes in the credibility mass of all declarations (small errors). Scenarios 4–6 assume significant changes in the mass of all declarations (large errors).

### 7.3. Calculation Results for Four Proportional Conflict Redistribution Rules

#### 7.3.1. The PCR5 Rule for 2 BBAs

The simulation results of identification information fusion using the PCR5 rule for two BBAs for deterministic scenarios 1–6 are presented in Figures 63–68.

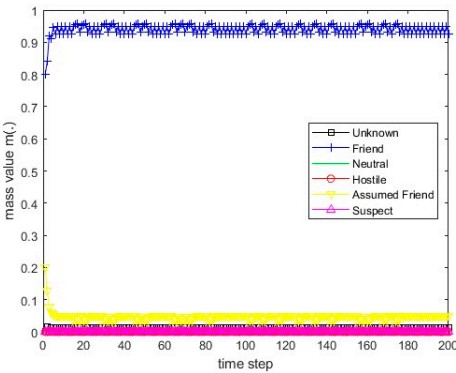

**Figure 63.** The values of the resulting belief mass for scenario 1 and the PCR5 rule for 2 BBAs.

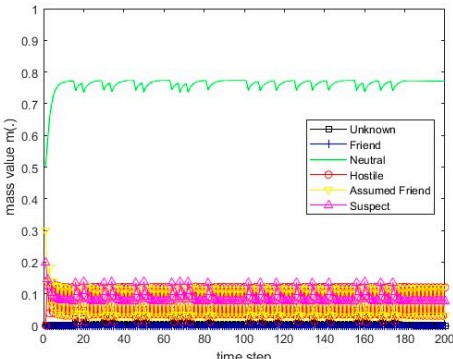

**Figure 64.** The values of the resulting belief mass for scenario 2 and the PCR5 rule for 2 BBAs.

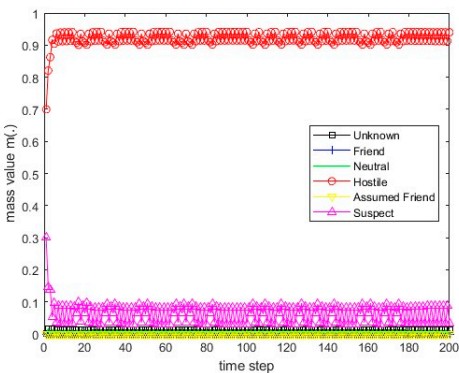

**Figure 65.** The values of the resulting belief mass for scenario 3 and the PCR5 rule for 2 BBAs.

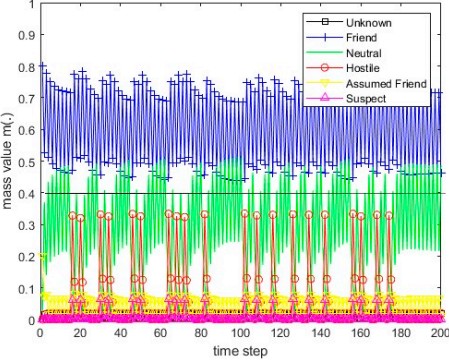

**Figure 66.** The values of the resulting belief mass for scenario 4 and the PCR5 rule for 2 BBAs.

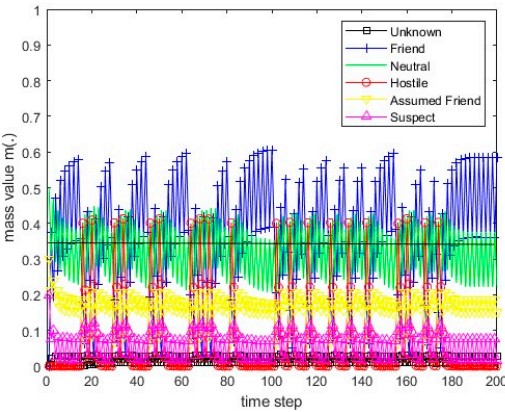

**Figure 67.** The values of the resulting belief mass for scenario 5 and the PCR5 rule for 2 BBAs.

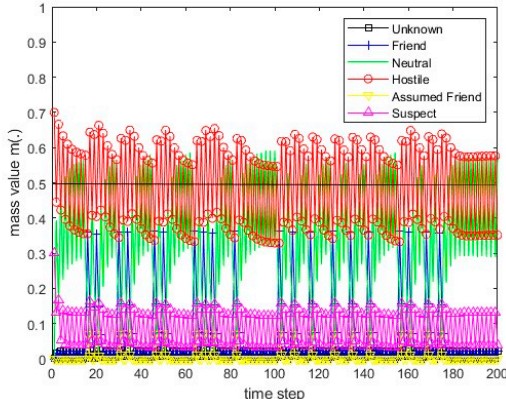

**Figure 68.** The values of the resulting belief mass for scenario 6 and the PCR5 rule for 2 BBAs.

For the PCR5 rule for two BBAs, the application of the decision thresholds at the belief mass level $m_\alpha = 0.40$ for scenario 4, $m_\alpha = 0.36$ for scenario 5, and $m_\alpha = 0.5$ for scenario 6 allows for a proper evaluation of the identification of the recognized object for most time moments.

### 7.3.2. The PCR5 Rule for 3 BBAs

The simulation results of identification information fusion using the PCR5 rule for three BBAs for deterministic scenarios 1–6 are presented in Figures 69–74.

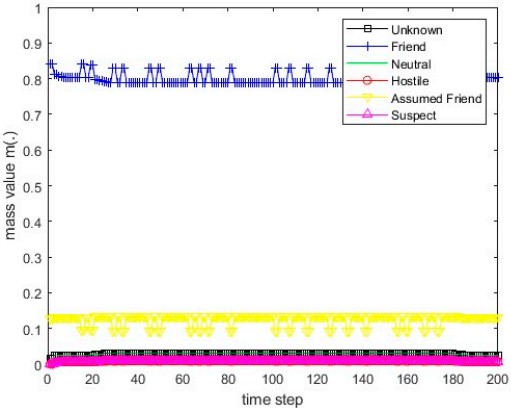

**Figure 69.** The values of the resulting belief mass for scenario 1 and the PCR5 rule for 3 BBAs.

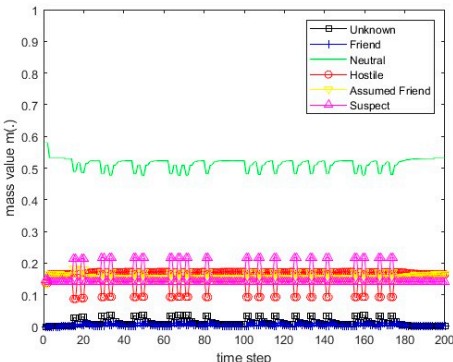

**Figure 70.** The values of the resulting belief mass for scenario 2 and the PCR5 rule for 3 BBAs.

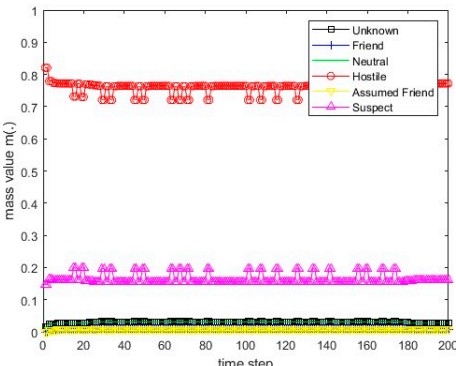

**Figure 71.** The values of the resulting belief mass for scenario 3 and the PCR5 rule for 3 BBAs.

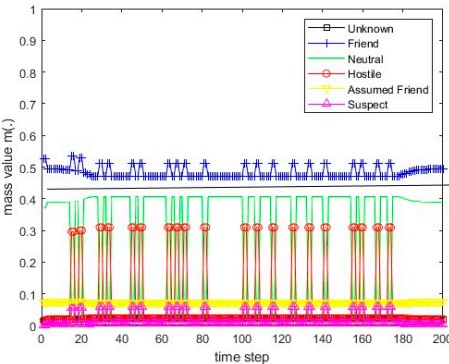

**Figure 72.** The values of the resulting belief mass for scenario 4 and the PCR5 rule for 3 BBAs.

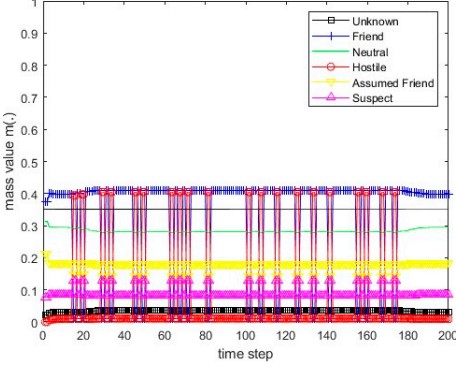

**Figure 73.** The values of the resulting belief mass for scenario 5 and the PCR5 rule for 3 BBAs.

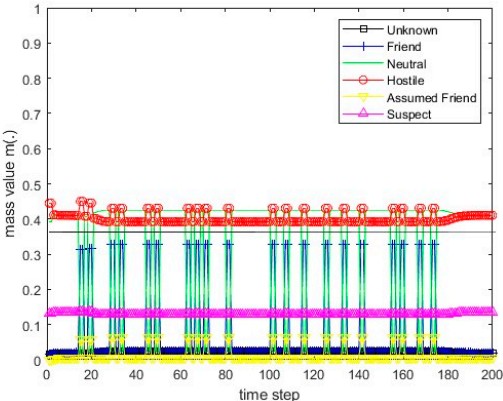

**Figure 74.** The values of the resulting belief mass for scenario 6 and the PCR5 rule for 3 BBAs.

For the PCR5 rule for three BBAs, the application of the decision thresholds at the belief mass level $m_\alpha = 0.42$ for scenario 4 allows for a proper evaluation of the identification of the recognized object.

For the PCR5 rule for three BBAs, the application of the decision thresholds at the belief mass level $m_\alpha = 0.37$ for scenarios 5 and 6 allows for a proper evaluation of the identification of the recognized object for most time moments.

### 7.3.3. The PCR6 Rule for 3 BBAs

The simulation results of identification information fusion using the PCR6 rule for three BBAs for deterministic scenarios 1–6 are presented in Figures 75–80.

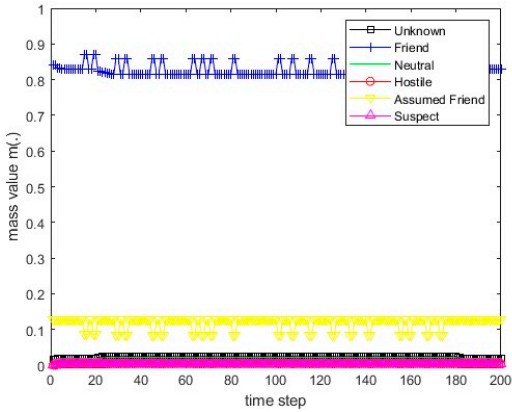

**Figure 75.** The values of the resulting belief mass for scenario 1 and the PCR6 rule for 3 BBAs.

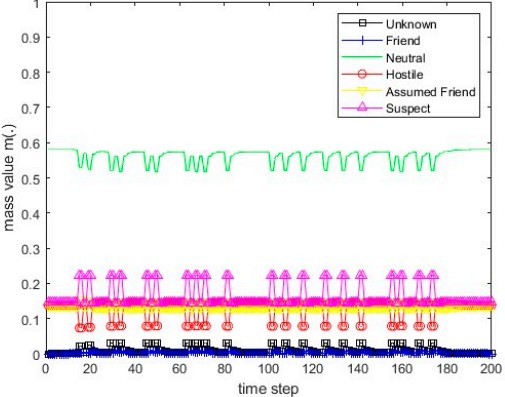

**Figure 76.** The values of the resulting belief mass for scenario 2 and the PCR6 rule for 3 BBAs.

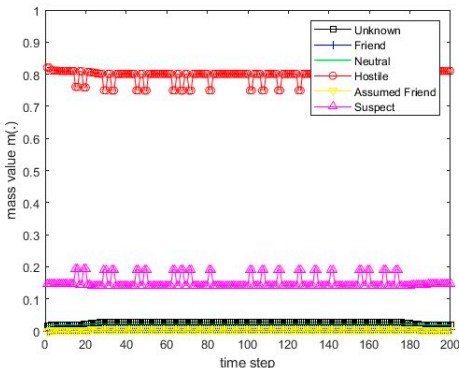

**Figure 77.** The values of the resulting belief mass for scenario 3 and the PCR6 rule for 3 BBAs.

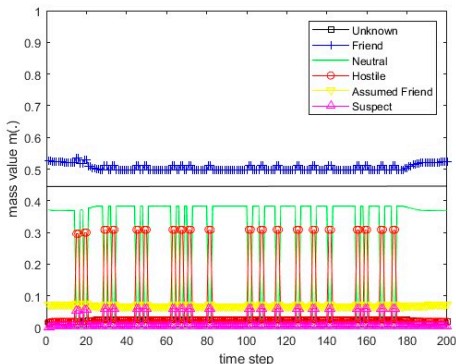

**Figure 78.** The values of the resulting belief mass for scenario 4 and the PCR6 rule for 3 BBAs.

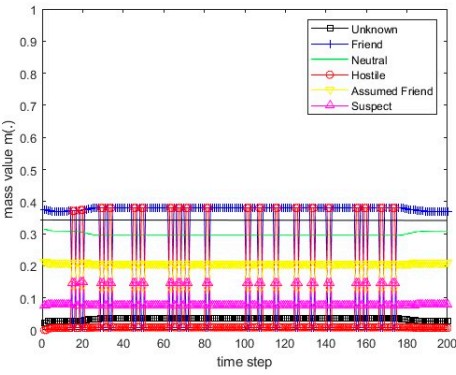

**Figure 79.** The values of the resulting belief mass for scenario 5 and the PCR6 rule for 3 BBAs.

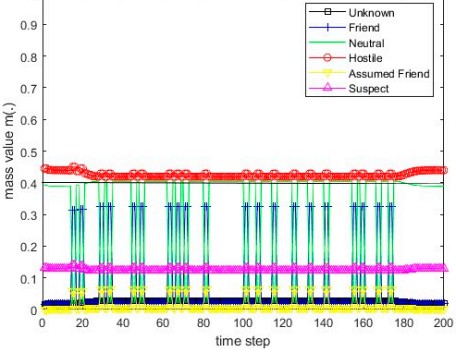

**Figure 80.** The values of the resulting belief mass for scenario 6 and the PCR6 rule for 3 BBAs.

For the PCR6 rule for three BBAs, the application of the decision thresholds at the belief mass level $m_\alpha = 0.45$ for scenario 4 allows for a proper evaluation of the identification of the recognized object.

For the PCR6 rule for three BBAs, the application of the decision thresholds at the belief mass level $m_\alpha = 0.35$ for scenario 5 and $m_\alpha = 0.4$ for scenario 6 allows for a proper evaluation of the identification of the recognized object for most time moments.

Comparing Figures 66–68 with Figures 72–74 and 78–80, conclusion can be drawn that the PCR5 for three BBAs and PCR6 for three BBAs rules provide more stable results of combined belief masses (smaller amplitude of changes). Due to the large dispersion of belief mass changes for scenarios 5 and 6, it is not possible to correctly evaluate the identification of the recognized object for all time moments.

The presented results (Figures 61–78) allow a conclusion to be drawn that the applied methods of removing conflicts in information fusion enables the correct conclusions to be drawn about the real identification of the recognized object.

## 8. Conclusions

The proposed basic belief assignment model for ESM sensors and radars can be used to build identification information-fusion systems. Models conformable to STANAG 1241 have primary practical significance.

Due to the assumption of conflicts between the ESM sensor declarations in this work, Dezert–Smarandache theory is used to determine the basic belief assignment of declarations as a product of the process of fusion of identification information sent by these sensors. Supplementing standard reports on the detected signals with random identification declarations allows the use of methods of identification information fusion in the information-fusion center. The test results confirm the full usefulness of conflict redistribution rules in reports from ESM sensors developed as a part of Dezert–Smarandache theory, with the best results presented for the PCR5 and PCR6 rule.

The basic belief assignment model for ESM sensors and for combined primary and secondary radars [17] can be applied to build models of different identification data-fusion systems. All the models compatible with STANAG 1241 have primary practical significance as it contains definitions corresponding to intersections of basic identification declarations. Therefore, the paper uses Dezert–Smarandache theory for the calculation of the basic belief assignment.

The conducted research showed that the best results were obtained for the PCR6 rule when reports from three sources (from two sensors and the fusion-system database) were processed simultaneously. This corresponds to the synchronous processing of reports and involves delayed processing of a report from one of the sources. The research confirmed a slight advantage of the PCR6 rule over the PCR5 rule. This was mainly the case when the sensors sent information with a high degree of conflict.

**Funding:** This work was financed by the Military University of Technology under Research Project UGB 866.

**Data Availability Statement:** Presented data accessible for authorised staff accordingly to local regulation.

**Conflicts of Interest:** The author declares no conflict of interest.

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
