# Peer review of "Fusion of Identification Information from ESM Sensors and Radars Using Dezert–Smarandache Theory Rules"

_remotesensing, doi:10.3390/rs15163977_

Round 1

Reviewer 1 Report

This article presents a method of fusion of identification information provided by combined primary and secondary surveillance radars and electronic support measures. The manuscript also describes the detail of proposed method clearly. And the evaluation also shows the improvement on different scenarios. This is a mature and comprehensive article.

There is only one question:

Some formulas (8,12,15,22…) end with "=" on the first line and start with "=" on the next line. Is this a typing error?

Minor editing of English language required

Author Response

Dear Reviewer,
I am pleased to resubmit a corrected manuscript entitled “Fusion of identification information from ESM sensors and radars using Dezert-Smarandache theory rules” and my response to your comments.

Reviewer 2 Report

The work addresses sensor data information fusion in the Dezert-Smarandache theory. The BBA in D-S theory is adopted for uncertain information modeling. The topic is important and the idea is interesting. I suggest a minor revision of your manuscript by considering the following comments.

1.      Please improve the logic in the ‘Abstract’. Currently, there are many so called ‘new ideas’ in sentences like ‘The paper presents…’ ‘It was proposed that…’ ‘The paper also presents…’

Instead, I suggest the authors describe the proposed method in a single uniform framework.

2.      Please improve Section ‘1. Introduction’. The logic is not that clear. E.g., there are two paragraphs that have only one sentence.

3.      The gap between the work and the previous ones should be based on many more works related to Dempster’s rule and its improvements.

4.      The literature review is not based on newly published works. Papers published in 2022/2023 are suggested such as “An Improved Failure Mode and Effects Analysis Method Using Belief Jensen–Shannon Divergence and Entropy Measure in the Evidence Theory. Arab J Sci Eng 48, 7163–7176 (2023). https://doi.org/10.1007/s13369-022-07560-4” or "A new correlation coefficient of mass function in evidence theory and its application in fault diagnosis". https://link.springer.com/article/10.1007/s10489-021-02797-2. Just to give an example.

5.      What is the relationship among Sections 3, 4, and 5? I suggest the authors give a uniform framework to describe the main contribution of the work in a flow chart or a separate section that can cover the contribution in Sections 3, 4, and 5.

6.      Some sentences need improvement. E.g., ‘rule have been presented for the deterministic scenarios 1 and 3 in Figures 12 and …’ in line 604 is not complete.

Moderate editing of English language required

Author Response

(The authors gave the same response as above.)

Reviewer 3 Report

This paper presents a method of fusion of identification (attribute) information provided by two types of sensors: combined primary and secondary (IFF) surveillance radars and ESM (Electronic Support Measures). This work is interesting, but it still needs further improvement. I provide some suggestions below for the revised version:

1. Please introduce research background, current situation, and existing problems in the introduction section in more detail.

2. The author should indicate the motivation and contribution of the article.

3. Page 4, line 150: “” should be “”.

4. Page 5, line 172: “Figure 1.” should be “Figure 2.”.

5. Page 7: Lack of corresponding references when introducing PCR1-PCR6.

6. Page 8, line 258: Missing subject for “is the non-zero sum of…”.

7. Page 14, line 359: “where Pd is the…” should be “where Pd is the…”.

8. Page 14, line 379: “Table 1. Transformation of the base belief… assignment mass” is redundant.

9. The references in the article are outdated. Please introduce some current related works in terms of information fusion based on evidence theory, e.g., Negation of the quantum mass function for multisource quantum information fusion with its application to pattern classification; GEJS: A generalized evidential divergence measure for multisource information fusion; A complex weighted discounting multisource information fusion with its application in pattern classification; Quantum X-entropy in generalized quantum evidence theory.

In short, I recommend accepting this paper after completing these modifications.

Minor editing of English language required. 

Author Response

(The authors gave the same response as above.)

Reviewer 4 Report

‘identification information from ESM sensors’ – ESM must be also written first in full. Then it is: ‘ESM (electronic intelligence - electronic support measures) electronic surveillance’. There are additional instances. E.g., Proportional Conflict Redistribution is written in full but some passages above it is listed as acronym. The DezertSmarandach theory (DSmT) is then written in full throughout the manuscript. Check such cases throughout the manuscript. Several statements made in the paper are not supported by adequate empirical evidence or by making reference to relevant literature. You should compare your results with others in terms of concrete data for better research integrative value. Replace it/they with the proper words to avoid confusion. E.g., ‘It leads to the Bayesian model of the basic belief assignment.’ A lot of ideas are either not substantiated, or only one source, typically old, is used. The manuscript has a low value of integration in the current debates on the topic: most cited sources are old and not from peer reviewed journals. The figures need more explanations. More development and depth of the methodology and analysis are needed. ‘rule have been presented for the deterministic scenarios 1 and 3 in Figures 12 and 13.’ - ? There is some discussion of the limitations of the study however these are not considered in terms of the implications on the study findings. The main contributions of the paper should be presented as part of the empirical discussions or critical assessment on the core research outcomes. The study lacks methodological rigor and the interpretation of results is not supported by adequate empirical evidence.

Author Response

(The authors gave the same response as above.)

Round 2

Reviewer 2 Report

'Figures 2 i 3' in Figure 4 can be improved.

Reviewer 4 Report

This revised version can be published.